# River flow in the near future: a global perspective in the context of a high-emission climate change scenario

Omar V. Müller[1,2], Patrick C. McGuire[3], Pier Luigi Vidale[3], and Ed Hawkins[3]

[1]Consejo Nacional de Investigaciones Científicas y Técnicas (CONICET), Santa Fe, Argentina.
[2]Centro de Variabilidad y Cambio Climático (CEVARCAM), Facultad de Ingeniería y Ciencias Hídricas (FICH), Universidad Nacional del Litoral (UNL), Santa Fe, Argentina.
[3]Department of Meteorology, National Centre for Atmospheric Science, University of Reading, Reading, United Kingdom

**Correspondence:** Omar V. Müller (ovmuller@unl.edu.ar)

**Abstract.** There is high confidence that global warming intensifies all components of the global water cycle. This work investigates the possible effects of the global warming on river flows worldwide in the coming decades. We conducted 18 global hydrological simulations to assess how river flows are projected to change in the near future (2015-2050) compared to the recent past (1950-2014). The simulations are forced by runoff from HighResMIP-CMIP6 GCMs, which assume a high-emission scenario for the projections. The assessment includes estimating the signal-to-noise (S/N) ratio and the time of emergence (ToE) of all the rivers in the world. Consistent with the water cycle intensification, the hydrological simulations project a clear positive global river discharge trend from ∼2000, that emerges beyond the levels of natural variability and becomes 'unfamiliar' by 2017 and 'unusual' by 2033. Simulations agree that the climate change signal is dominated by strong increases in flows of rivers originating in Central Africa and South Asia, and those discharging into the Arctic Ocean, partially compensated by the reduced flow projected for Patagonian rivers. The potential implications of such changes may include more frequent floods in Central African and South Asian rivers, driven by the projected magnification of the annual cycles with unprecedented peaks, a freshening of the Arctic Ocean from extra freshwater release, and limited water availability in Patagonia given the projected drier conditions of its rivers. This underscores the critical need for a paradigm shift in prioritizing water-related concerns, amidst the challenges of global warming.

## 1 Introduction

Rivers play a vital role in the Earth System, being essential for the global water cycle, habitat, transport, agriculture, and energy. At the same time, under anomalous conditions, rivers may cause devastating damage through floods or by limiting navigability and water abstraction. As an integrator of the water balance over land, river flow is sensitive to changes in precipitation, evapotranspiration, and soil moisture. Shifts in regional precipitation amount, intensity, and patterns, and/or in the interplay between soil moisture and evapotranspiration regimes, may produce anomalous river flow. The magnitude of the anomaly will depend on the type of catchment and the intensity of the change.

There is high confidence that global warming has modified all components of the global water cycle in recent decades (Caretta et al., 2022). The observed changes vary from regions with increased mean and extreme land precipitation to regions

with reduced precipitation, or even zones with heavier precipitation events, separated by longer dry spells. Evapotranspiration has changed in response to changes in precipitation and warmer temperatures, as well as to the observed vegetation greening in northern high latitudes (Yang et al., 2023), altering the ability of the soil to hold moisture. Moreover, higher temperatures directly alter snow accumulation and ablation processes causing shrinking of mountain glaciers, land ice, and snow cover. All these changes directly affect runoff generation, and thereby river flow variability, and even river flow trends. Dai et al. (2009) reported significant (both positive and negative) trends in 55 large rivers during 1948–2012. Alkama et al. (2011) reinforces the notion that runoff trends are a regional scale issue. They attribute these trends to precipitation variability while also emphasizing the potential impact of human-induced global warming on high-latitude river discharge, specifically through its effect on permafrost and glaciers. Similarly, Gudmundsson et al. (2021) reported heterogeneous trend patterns across the world in low, mean, and high flow, with some rivers drying and others wetting during 1971-2010.

The continuation of global warming is projected to intensify the exchanges of water between the land, the ocean, and the atmosphere (Alkama et al., 2013; Douville et al., 2021). In all scenarios, the CMIP6 multi-model ensemble projects an overall increase in mean and extreme land precipitation, albeit with substantial variations across regions. Projected changes in evapotranspiration and soil moisture remain uncertain, as they are not only modulated by meteorological changes, but also by plant acclimation to higher $CO_2$ (Lemordant and Gentine, 2019; Oliver et al., 2022). This uncertainty extends to runoff, and by that, to river flow, which is the local runoff that is subsequently routed from land to oceans through river channels. Douville et al. (2021) and Zhou et al. (2023) concur on the projected increase in global runoff in the coming decades, albeit attributing it to different factors. Douville et al. (2021) link this rise to global warming, with confidence levels escalating with emissions scenarios. In contrast, Zhou et al. (2023) attribute it to changes in the synergistic effects of vegetation responses to rising CO2 concentrations and land surface reactions to radiative changes, which lead to a shift in precipitation partitioning towards runoff instead of evapotranspiration.

Considering the observed and the projected changes in global runoff and knowing their strong regional variability, it is relevant to explore how runoff changes alter the flow of all rivers of the world. A first approach is to quantify the magnitude of the changes in river flow (e.g., Nijssen et al. 2001; Koirala et al. 2014; Döll et al. 2018; Gudmundsson et al. 2021). However, an extra step that enhance such standard analysis is to locally determine the signal-to-noise (S/N) ratio of any changes and estimate the time of emergence (ToE). These concepts, initially used in the IPCC AR4 (Christensen et al., 2007), indicate where and when a climate change signal emerges from the background natural variability, i.e., where and when climate change might start having larger impacts (Hawkins et al., 2020). The ToE methods are often applied to temperature (e.g., Mahlstein et al. 2011; Hawkins and Sutton 2012; Mora et al. 2013) and precipitation (e.g., Giorgi and Bi 2009; Mahlstein et al. 2012; Hawkins et al. 2020), albeit rarely for other variables. Some exceptions are Lyu et al. (2014) who estimated the ToE for sea-level in a global study, or Muelchi et al. (2021) who calculated the ToE for runoff in Switzerland. Given that changes in rivers due to changing climate have potentially far reaching implications for human populations (Nijssen et al., 2001), further research about their evolution and their ToE is expected to provide valuable information for impact and adaptation studies.

The main purpose of this paper is to provide an insight of the possible effects of global warming in a high-emission scenario on river flows, at the global scale, over the next few decades. In order to fulfil this objective, we simulate rivers worldwide by

forcing a river routing model with runoff from CMIP6 GCMs, validate them, evaluate their projected anomalies, and calculate their ToE. The regions of the world presenting stronger signal of climate change are further explored to infer the potential impacts of such changes. The study is organized as follows: Sect. 2 describes the used GCMs, the hydrological model, and the river flow assessment methodology; Sect. 3 presents the validation of the hydrological simulations, examines projected river flow changes and their ToE with focus on the regions of the world that are projected to change the most; and Sect. 4 presents a discussion of the results and summarizes the concluding remarks.

## 2 Data and methods

### 2.1 GCM simulations and river routing model

A set of 18 GCM simulations (see Table 1), produced within the framework of the High Resolution Model Intercomparison Project (HighResMIP v1.0) for CMIP6 (Haarsma et al., 2016), force the river routing model used to evaluate the rivers in the near future. The selection of these HighResMIP GCMs was contingent upon the availability of surface and subsurface runoff data within the Earth System Grid Federation (ESGF) servers, which host the HighResMIP simulations. The simulations include five different GCM families: CNRM-CM6 (Decharme et al., 2019; Voldoire et al., 2019), EC-Earth3P (Haarsma et al., 2020), HadGEM-GC31 (Williams et al., 2018), MRI-AGCM3-2 (Mizuta et al., 2012), and NICAM16 (Kodama et al., 2021), which vary in the simulation type and the horizontal resolution. The simulation type can either be atmosphere-land (AMIP) or ocean-atmosphere-land (COUPLED). All GCMs present AMIP simulations, but just CNRM-CM6, EC-Earth3P, and HadGEM-GC31 have COUPLED simulations. In addition, all GCMs produced a low- and a high-resolution simulation, except for the HadGEM-GC31 family that also provides at intermediate-resolution. To reconcile the variety of grid topologies used by the different GCMs (rectilinear, reduced gaussian, icosahedral, etc) in the comparison of the GCMs' resolution, we provide the atmospheric horizontal resolution at $50°$N. This mid-latitude serves as a representative point for assessing resolution, particularly given the significant variation in resolution from the Equator to the poles in models using rectilinear grids. The atmospheric resolution at $50°$N ranges from $25\,\mathrm{km}$ to $134\,\mathrm{km}$ for the set of simulations (Table 1). For COUPLED simulations we also provide the ocean resolution, which varies from $1\,°$ for low-resolution to $0.25\,°$ for high-resolution simulations. Note that there is only one member per resolution and simulation type, which may limit the robustness of this study, because of internal climate variability. To shed light on this, we perform a comprehensive internal variability analysis of a set of 58 individual realizations across different GCMs in Appendix A. The results show that the inter-model variability is much larger than the internal variability, which suggests strong robustness of the set of simulations used in this study.

The total runoff (surface and subsurface) produced by each GCM simulation is used to force the TRIPpy (Total Runoff Integrating Pathways in python) river routing model (Müller, 2023), a standalone implementation of the original TRIP model developed by Oki and Sud (1998) in FORTRAN. TRIPpy collects runoff from each grid cell and drives it through the river network to estimate the river storage and outflow of each grid cell. The simulations are run globally (excluding Antarctica) using nearest-neighbour to regrid the runoff from the original GCM resolutions to the target grid at a common resolution

**Table 1.** GCM simulations.

| GCM | Simulation type AMIP, COUPLED | Atmosphere horizontal resolution at 50° [km] | Ocean resolution for COUPLED [deg] | Warming [°C] AMIP, COUPLED |
|---|---|---|---|---|
| CNRM-CM6-1 | yes, yes | 100 | 1.00 | 1.2, 1.4 |
| CNRM-CM6-1-HR | yes, yes | 35 | 0.25 | 1.2, 1.3 |
| EC-Earth3P | yes, yes | 80 | 1.00 | 1.1, 1.4 |
| EC-Earth3P-HR | yes, yes | 39 | 0.25 | 1.1, 1.3 |
| HadGEM-GC31-L* | yes, yes | 134 | 1.00 | 1.2, 2.0 |
| HadGEM-GC31-MM | yes, yes | 60 | 0.25 | 1.2, 1.7 |
| HadGEM-GC31-HM | yes, yes | 25 | 0.25 | 1.2, 1.8 |
| MRI-AGCM3-2-H | yes, no | 60 | —- | 1.2, —- |
| MRI-AGCM3-2-S | yes, no | 20 | —- | 1.2, —- |
| NICAM16-7S | yes, no | 56 | —- | 1.1, —- |
| NICAM16-8S | yes, no | 28 | —- | 1.1, —- |

*=M for AMIP (HadGEM-GC31-LM) and

*=L for COUPLED (HadGEM-GC31-LL)

of 0.25°. The quarter-degree river network is based on the flow direction of the Dominant River Tracing dataset (Wu et al., 2011, 2012).

TRIPpy employs a simple advection method within a water balance model to route total runoff through the topography. This method calculates changes in river channel storage within each grid cell by accounting for the difference between the inflow, which includes both local runoff and contributions from upstream grid cells, and the outflow. The outflow is estimated using a linear function of storage, considering the river flow velocity and the river length between two connected grid cells. Detailed TRIPpy equations can be found in the appendix of Müller et al. (2021a), who evaluated simulated river flow across 334 monitored catchments, revealing promising performance compared to observed data. The model's key attribute lies in its simplicity, enabling long-term global simulations with minimal computational resources, all while delivering commendable performance.

The hydrological simulations span from 1950 to 2050 at monthly time-scale, considering 1950-2014 as the present climatology (hereinafter PRESENT), and 2015-2050 as the near future (hereinafter FUTURE). Note that the projections in High-ResMIP consider a scenario as close to CMIP5 RCP8.5 as possible within CMIP6 (Haarsma et al., 2016), i.e., the hydrological predictions are appraised in the context of a high emission scenario. Table 1 indicates the change in global temperature between FUTURE and PRESENT as an indicator of the assumed scenario impact on the projections. AMIP projections present a warming of ∼1.2 °C, while COUPLED projections present a change ranging from 1.3 °C to 2 °C. Although such changes may seem large for short-term climate predictions (36 years), they are likely to occur in the long-term.

To ensure the robustness of both the model and the forcing GCMs for our specific objectives, we undertake a comprehensive validation of the 18 hydrological simulations and the 18-model ensemble mean simulation for the PRESENT period. The validation involves assessing four metrics which evaluate diverse aspects of the simulations, computed by comparing our simulations with monthly observations of 346 selected near-coast gauge stations from Dai (2017) dataset. The selection criteria for observations focused on data availability (>120 observed values for the PRESENT period), a minimum size of the catchment (>14 grid cells), and an agreement between catchment area in model and observations (>65 %). The selected monitored rivers cover approximately 42% of the global land and contributes to about 45% of the global river discharge (see Fig. 1a). The validation metrics are:

- Relative Bias ($RB$): Measures the percentage difference between total simulated and observed mean flow for all monitored rivers, indicating whether simulations tend to overestimate or underestimate river flows. Range: $[-100, \infty)$, Perfect score: 0.

- Overlapping Coefficient ($OC$): Quantifies the overlapping area below the curves of the two distributions, reflecting the degree of agreement between them. Range: $[0, 1]$, Perfect score: 1 (Weitzman, 1970; Müller et al., 2021a).

- Correlation Coefficient ($r$): Indicates the degree of linear relationship between modeled and observed values. Range: $[-1, 1]$, Perfect score: 1.

- Non-parametric Kling-Gupta Efficiency ($npKGE$): Considers errors in the mean (evaluated with the ratio between model and observed mean values), variability (measured with the normalized flow–duration curve), and dynamics (evaluated with the Spearman rank correlation). Range: $(-\infty, 1]$, Perfect score: 1 (Pool et al., 2018).

## 2.2 Assessment methodology

To understand the projected changes in rivers in the next decades, we perform a three-steps analysis. First, we identify the main differences between FUTURE vs PRESENT in key hydrological variables. Second, we estimate the ToE of river discharge worldwide. Lastly, we focus the evaluation of river flow on regions exhibiting notable anomalies and where there is a clear consensus among model simulations regarding the projections.

In the comparison of FUTURE vs PRESENT we assess the projected anomalies of land precipitation and total runoff in the near future (2015-2050) with respect to the recent past (1950-2014), used as reference climatology. A particular interest is given to the level of agreement among GCMs in such changes, which ensure robustness to the climate change signal (if any). Then, we centre the analysis in river flow to understand how anomalies in runoff end up affecting the different rivers of the world.

The river discharge S/N ratio and ToE is calculated following the approach proposed by Hawkins and Sutton (2012). The goal of the method is to decouple the climate change signal ($S$) from the natural variability (the noise $N$). In our work, the method is applied to the river discharge annual anomaly ($Q$) of each simulation. PRESENT is used as the base-period to calculate the anomalies in the entire period (1950-2050).

At the global scale, the signal $S_G(t)$ is a low-pass filtered version of the original $Q_G$ time-series. The filter is based on the convolution of a scaled window of 41-year length with the signal, resulting in a smoothing effect of the inter-annual variability. On the other hand, the noise is a fixed value calculated as $N_G = \sigma(Q_G(t) - S_G(t))$ over the base-period, where $\sigma$ is the standard deviation. At the local scale (grid box), the signal is a linear regression of the local river flow annual anomaly $Q_L(t)$ with respect to the global signal $S_G(t)$, that is, $S_L(t) = mS_G(t) + b$, where $m$ and $b$ are the regression coefficients (slope and intercept respectively). The local noise is then estimated similarly to the global case as $N_L = \sigma(Q_L(t) - S_L(t))$ over the base-period. The logic behind the methodological decisions (e.g., choice of filter, linear regression) results from a comprehensive analysis summarized in Appendix B.

Both scales (global and local) use the corresponding noise as a threshold to determine the year in which the signal of climate change emerges from the natural variability. Following the terminology used by Frame et al. (2017) and Hawkins et al. (2020), the year $t$ in which $|S(t)| > N$ is described as ToE to 'unfamiliar' climate, while the year in which $|S(t)| > 2N$ as ToE to 'unusual' climate conditions. Conversely, $|S(t)| < N$ means that the projections of river flow remain in the range of its historical variability.

The previous analysis enables the identification of regions where river flow predictions (1) indicate a transition from their established climate to an unfamiliar or even unusual climate, and (2) exhibit a significant consensus among models. Further assessment of these regions aims to determine the timing of the shift and its potential impact.

## 3   Results

### 3.1   Validation of Hydrological Simulations

The evaluation of the high-resolution hydrological simulations across 346 monitored catchments provides a nuanced understanding of model performance. These results not only offer valuable insights into the models' consistency but also underscore their effectiveness in capturing key hydrological features. The metrics employed, including $RB$, $OC$, $r$, and $npKGE$, offer distinct perspectives on the models' performance.

Figure 1b summarizes the scores for the set of simulations. Relative bias ($RB$) provides insights into the mean volume biases between simulated and observed flows, ranging from -4.2% to 27.7%. Overlapping coefficient ($OC$) assesses the agreement in flow volumes, while correlation ($r$) focuses on the ability of the models to capture observed river flow variability. Both scores range from 0.51 to 0.72. Lastly, non-parametric Kling-Gupta efficiency ($npKGE$), the most demanding metric, evaluates flow volume, variability, and dynamics, resulting in scores ranging from 0.39 to 0.58. Despite the diversity of these metrics, there is consistency across models, with all exhibiting values in a narrow range. Notably, the ensemble mean simulation outperforms most individual models, with $RB = 6.8$ %, $OC = 0.63$, $r = 0.72$, and $npKGE = 0.58$. Among the top-performing models are the GCMs of the EC-Earth3P and MRI-AGCM3-2 families.

When the assessment is restricted to the 20 largest monitored rivers most scores are increased, especially for $r$ and $npKGE$ (see Fig. 1c). For instance, for the ensemble mean simulation, this targeted analysis yields notable improvements, with $RB$ reducing to -1.5%, correlation enhancing to 0.76, and $npKGE$ rising to 0.71. These results suggest that GCMs demonstrate

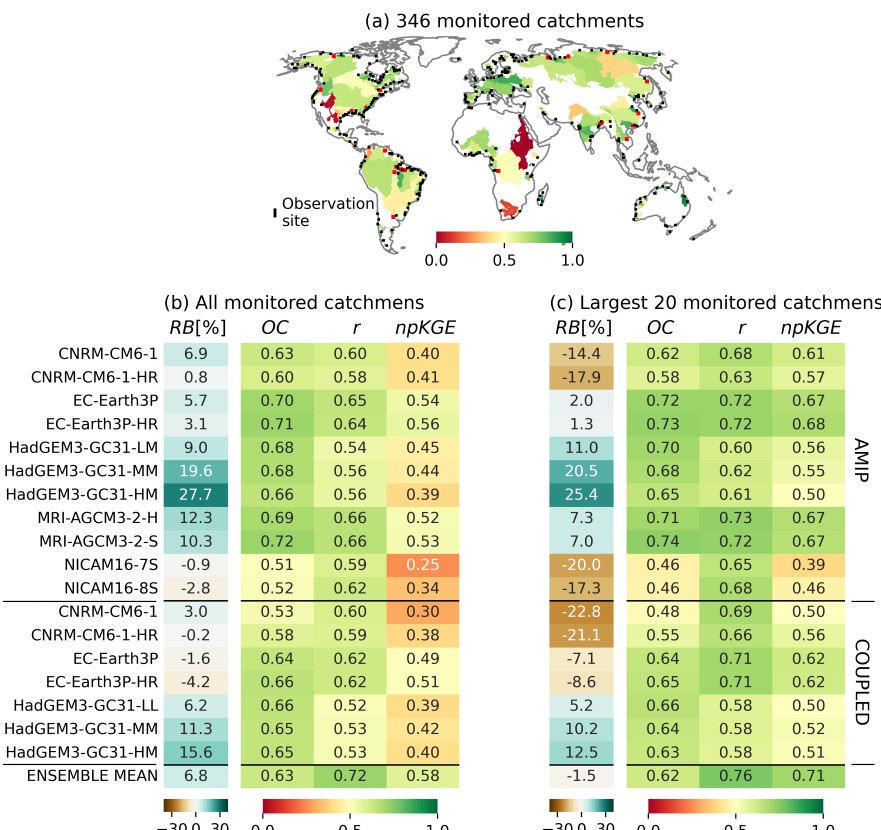

**Figure 1.** Validation of the global hydrological simulations. (a) Monitored rivers with black dots indicating the observation sites for river flow and colors showing the overlapping coefficient of the ensemble mean simulation for each monitored river. (b) Average relative bias ($RB$), overlapping coefficient ($OC$), correlation coefficient ($r$), and non-parametric Kling-Gupta Efficiency ($npKGE$) for each simulation and the ensemble mean simulation. (c) As (b) but for the 20 largest catchments. The averaged $OC$, $r$, and $npKGE$ are calculated using a weighted average, where the weight assigned to each river is proportional to its contribution to the total observed flow under evaluation.

a greater capacity to accurately simulate the variability and dynamics of runoff generation in large catchments, compared to smaller ones. The consistency in performance across models, independently of the resolution and the type of GCM forcing the hydrological simulations, and the enhancement in scores when focusing on major rivers collectively demonstrate commendable performance. This showcases the simulations' ability to reproduce observed large scale hydrological patterns, reinforcing our confidence in the models' suitability for our study objectives.

### 3.2 Changes in the land water budget

Precipitation, evapotranspiration, and runoff are the main components of the long-term land water budget and thereby, the key hydrological variables to understand long-term changes in rivers. Figure 2 compares how global mean values of these variables

change in the projections with respect to the climatology. Notably, all the models agree in the prediction of wetter conditions for the next decades independently of the type of simulation (AMIP or COUPLED) and the model's resolution. However, the positive changes in the projections of precipitation and runoff tend to be stronger for the the COUPLED simulations, likely due to the higher level of global warming they simulate for the FUTURE period (see Table 1). On the other hand, for each GCM family, increasing the resolution results in higher values for both the PRESENT and FUTURE periods. That is because high-resolution models enhance ocean-land moisture transport, producing more realistic mesoscale circulation patterns and synoptic systems (Vannière et al., 2019; Müller et al., 2021b). Moreover, the better-resolved orography at high-resolution favours the organization of convective precipitation and improves the representation of orographic jets producing more orographic precipitation, and thereby more runoff in the headwaters (Vellinga et al., 2016; de Souza Custodio et al., 2017; Vannière et al., 2019; Müller et al., 2021a). The differences that arise with resolution and the level of warming produces a spread in the global mean values of the various GCMs. Even so, it is noteworthy that the values for all GCMs remain in the range of the observational uncertainty for the three variables.

Despite the global land precipitation increases by $3.6 \times 10^3$ km$^3$yr$^{-1}$ in the ensemble mean, which represents just 3 % more precipitation, a large fraction of the extra water ends up in runoff, which is augmented by $2.4 \times 10^3$ km$^3$yr$^{-1}$, representing a positive change of 6 % in the global average. The remaining extra precipitation is returned to the atmosphere through evapotranspiration, which rises by 2 % in the ensemble mean. The global rise of land precipitation and evapotranspiration is mainly explained by two factors that have a general consensus of most GCMs: a strengthening of the Intertropical Convergence Zone (ITCZ) and an overall wettening in the northern high-latitudes (a discussion about such phenomena is given in Appendix C).

Positive anomalies in precipitation are amplified in runoff (in terms of percentage change) when the extra water either falls over wet regions, where there is no more room for evapotranspiration or, over mountainous areas, where horizontal fluxes prevail (Müller et al., 2021a). Figure 3 shows that positive and negative changes in runoff are unevenly distributed in the world. Central Africa is the most extensive region with strong wetter conditions, but also more runoff is predicted for southeast South America, India, the Maritime Continent, and the windward side of orographic barriers like Tropical Andes, Alaska Range, and the Himalayas. On the other hand, the main reductions of runoff are projected in parts of the Amazon Forest and southern Chile. There is agreement among most models on the regions presenting notable changes (either positive or negative), but also about the slight increase of runoff in the northern high latitudes, which is related to the strong signal of warming projected for that area (see Fig. C1c and its description in Appendix C).

The predicted global enhancement in runoff has direct effect on river flow. Figure 4a presents the percentage change in river discharge between FUTURE and PRESENT for the catchments of the world, while Fig. 4b depicts similar information but detailed for all rivers tributaries. Consistently with the analysis of runoff, the stronger positive changes appear in African, Australian, and Boreal rivers. In Africa, many important rivers increase the mean discharge by more than 20 %, including the three major rivers: Congo (+20 %), Nile (23 %), and Niger (26 %), but also Okavango (+21 %), Volta (+33 %), and rivers feeding Lake Chad, whose catchment presents the largest percentage increment (+49 %). In Australia, the major river, Murray, is augmented by 14 % while other small rivers in northern Australia (Victoria, Ord, Fitzroy) and those discharging into the

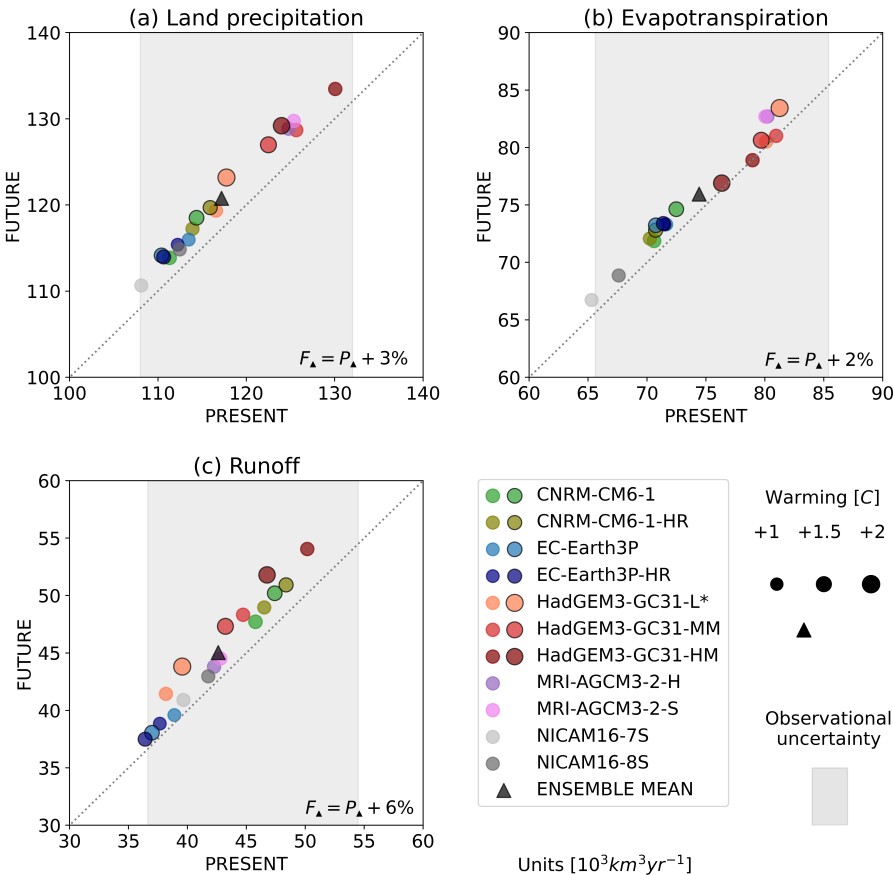

**Figure 2.** Scatterplots of the global land surface water budget components: (a) precipitation, (b) evapotranspiration, and (c) runoff comparing PRESENT mean vs FUTURE mean of each GCM simulation. The GCMs are shown with circles, using black border for COUPLED simulations, while the rest are for AMIP simulations. The ensemble mean based on the 18 simulations are represented with triangles. The markers' size is proportional to the degree of warming of each simulation (see values in Table 1). The legend in the bottom right corner of each scatterplot indicates the percentage change between FUTURE and PRESENT of the ensemble mean ($F_{\blacktriangle}$ and $P_{\blacktriangle}$ respectively). The * in the legend means M for AMIP (HadGEM3-GC31-LM) and L for COUPLED (HadGEM3-GC31-LL). The grey bands show the observational uncertainty considering a large number of observation-based estimations including: IPCC AR6 (Caretta et al., 2022), Rodell et al. (2015), Trenberth et al. (2007), ERA5 (Hersbach et al., 2020), CRU TS4.05 (Harris et al., 2020), WFDEI (Weedon et al., 2018), CPC (Chen et al., 2008), FLUXCOM (Jung et al., 2019), Dai et al. (2009), Clark et al. (2015), Müller et al. (2021a), and GLOFAS (Harrigan et al., 2020). Units are in $10^3 \mathrm{km}^3\mathrm{yr}^{-1}$.

Lake Eyre (Cooper, Warburton, among others) increase their flow by more than 35 %. In the boreal zone, almost all rivers simulate more drainage of freshwater into the Arctic Ocean, being those located in east Russia (e.g., Lena, Yana, Kolima), Alaska (e.g., Yukon) and Greenland the rivers with at least 10 % more freshwater. In South America, just the Uruguay river presents a significant rise of discharge (15 %). On the other hand, a few small rivers in the world present reduced flow for the

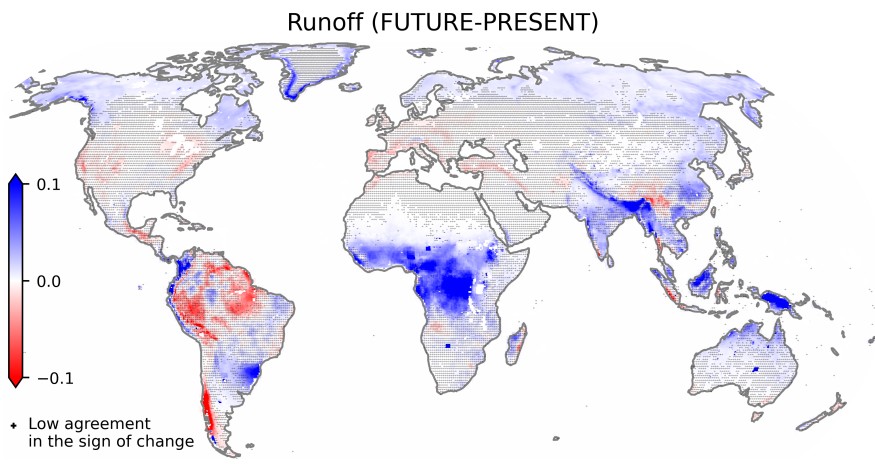

**Figure 3.** Multi-model ensemble mean differences in runoff $[10^3 \text{km}^3 \text{yr}^{-1}]$ between FUTURE (2015-2050) and PRESENT (1950-2014). The crosses indicate that at least 3 out of 18 GCMs disagree in the sign of change.

FUTURE. For instance, rivers originating in Southern Andes (e.g., Maipo, Maule, Limay, Negro, Chubut) and Colorado in the USA decrease their flows by $\sim$15 %, while most rivers of the Iberian Peninsula project a decay of about $\sim$10 %. Interestingly several southern tributaries of Amazon present dry anomalies, but they are not sufficient to significantly alter the downstream discharge into the Atlantic Ocean. In summary, a 6 % extra global runoff in the near future may seem irrelevant, but the changes are heterogeneously distributed throughout the globe, with many important rivers changing their mean flow by more than 15 %, which suggests a clear signal of climate change.

## 3.3 Time of emergence

Under the imposed high-emission scenario, all GCMs project a global rise of river discharge for the next decades, and there is an overall consensus among models on where the changes of river flow are likely to occur. However, there is an important spread in the magnitude of the change. Figure 4a presents the trends of global river discharge anomalies for each model. The differences among models get amplified over time and are more noticeable in COUPLED models, i.e., the stronger signal of change are simulated by the GCMs with higher warming (see Table 1). The ensemble mean global river discharge for PRESENT is $42.6 \times 10^3 \text{km}^3 \text{yr}^{-1}$, while the anomalies by 2050 are in the range $[0, 4.9] \times 10^3 \text{km}^3 \text{yr}^{-1}$ for AMIP and $[0.4, 8.1] \times 10^3 \text{km}^3 \text{yr}^{-1}$ for COUPLED. These anomalies represent a positive change of up to 11.5 % for AMIP and up to 19.0 % for COUPLED by the end of the projected period.

But, are these anomalies within the natural variability range? Figure 5b presents the ToE estimation for the ensemble mean. As for individual models, the ensemble mean presents a steady-state until about the year 2000, and a strong positive trend thenceforth. The anomalies remain within the natural variability range ($\pm N$), i.e. within the familiar climate conditions, until

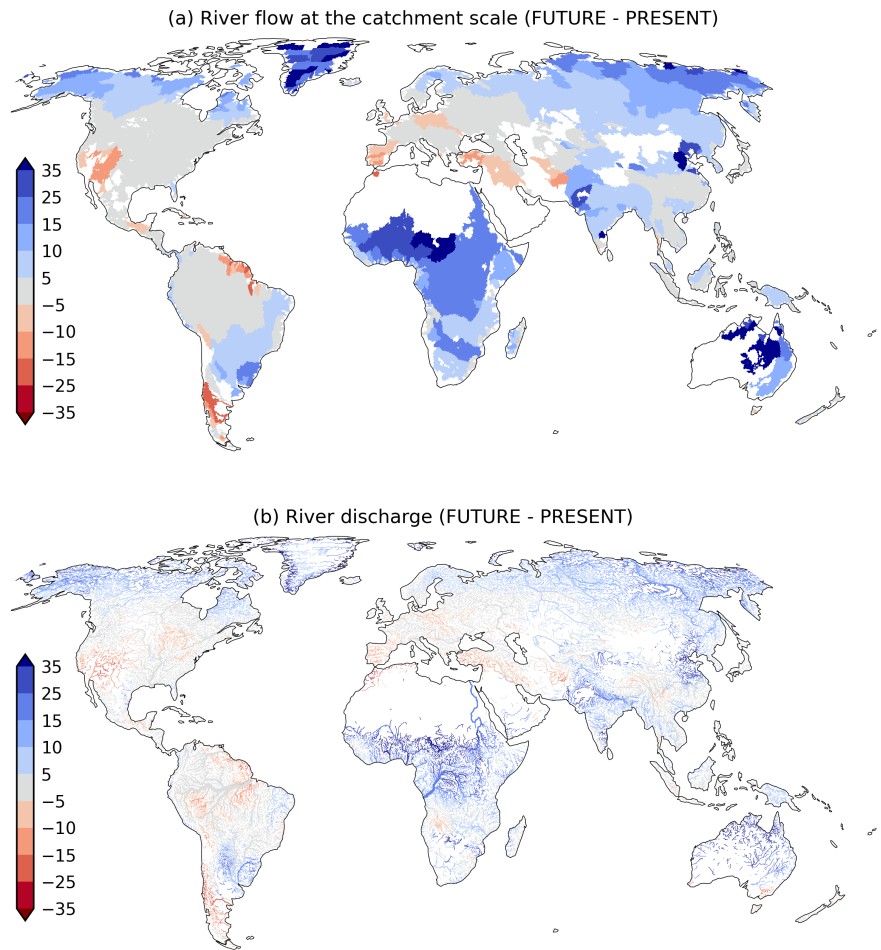

**Figure 4.** Multi-model ensemble mean differences in river flow between FUTURE (2015-2050) and PRESENT (1950-2014) presented as (a) the average difference at the catchment scale (i.e., the difference calculated at the river mouth of each catchment) and as (b) the difference for each channel of the river network. Rivers with little climatological flow ($< 100 \, \mathrm{m}^3\mathrm{s}^{-1}$ at the river mouth for the top panel and $< 5 \, \mathrm{m}^3\mathrm{s}^{-1}$ in the river channel for the bottom panel) are masked out in the maps. Units are in %. The outstanding differences observed in Greenland and central Australia between Figs. 3 and 4a results from the mathematical magnification of percentage changes in areas with small mean values and the depiction of uniform runoff differences at the catchment scale driven by strong changes near deltas (in particular for Greenland).

the year 2017. From there on, the global river discharge enters in an unfamiliar climate until 2033, when it shifts to unusual climate condition.

The emergence of global river discharge can have severe implications for specific rivers around the world, such as an increased frequency of floods. Figure 6 displays the ensemble mean spatial distribution of the signal-to-noise ratio ($S/N$) by the year 2050, when the global signal is maximum (Figure 5b). The pattern reveals that the majority of rivers worldwide will remain in a range of natural variability in the coming decades ($|S/N| < 1$). However, most changes arise in high-latitude and

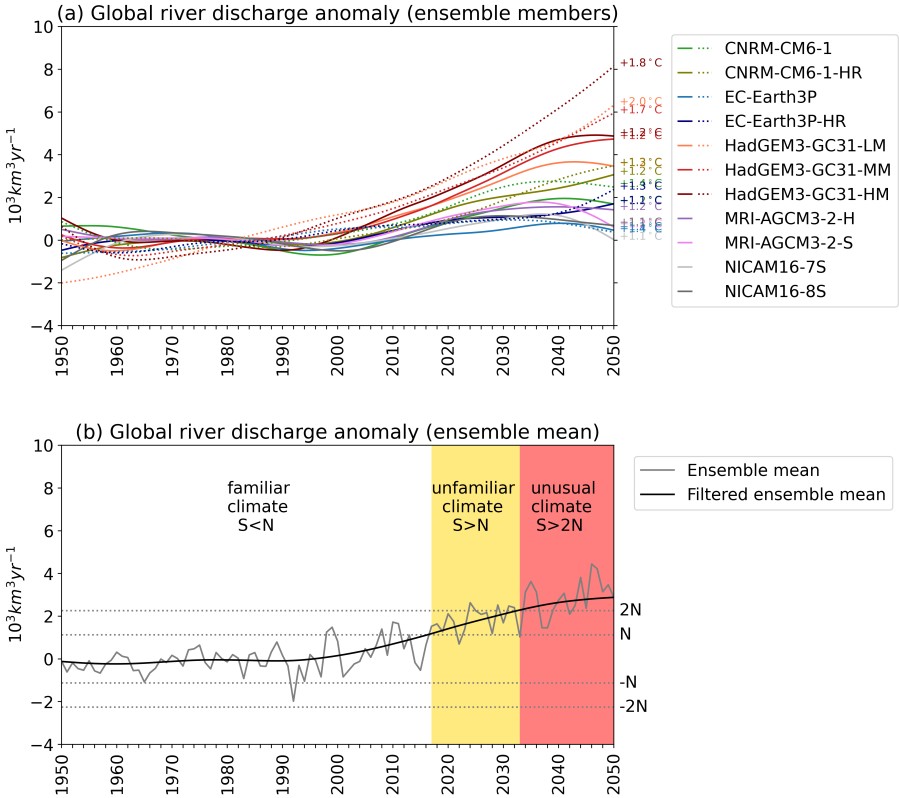

**Figure 5.** (a) Temporal evolution of global river discharge anomalies smoothed with a low-pass filter for each ensemble member. Solid lines are for AMIP and dashed for COUPLED simulations. Numbers on the right side indicates the average warming in the FUTURE period. (b) As (a) but for the ensemble mean in black, and the annual anomalies in grey. Dashed lines are thresholds to identify the year when the signal of climate change emerge from the natural variability ($N$) to unfamiliar (yellow) or unusual (red) climate conditions. In all cases the anomalies are calculated as the departure from the mean of the PRESENT period (1950-2014).

tropical areas where $|S/N| > 0.3$. The high-latitude changes can be attributed to polar amplification, while the tropical changes
are likely due to a shift to intense precipitation in the ITCZ (see discussion in appendix C), which is accurately simulated only
at resolutions finer than 20 km. In this sense, the river network may act as a strong filter, partly compensating for precipitation
errors. Within high-latitude and tropical areas, rivers originating in central Africa, east Russia, Alaska, and Greenland present
signals of climate change ($|S/N| > 1$). Figure 7a shows that the main courses of Congo, Nile, Niger, and Chad present a ToE
from familiar to unfamiliar climate during the years 2015-2025, while Yukon and Lena after 2030. Moreover, the projections of
250 river flow in lower Congo, Oubangui (Congo's north tributary), Chari (primary tributary of Lake Chad), and Main Nile indicate
a shift to unusual wetter climate condition during the 2030s (Fig. 7b). Similarly, the flow of rivers discharging in the Greenland
coasts are projected to move to unusual climate in the next decade.

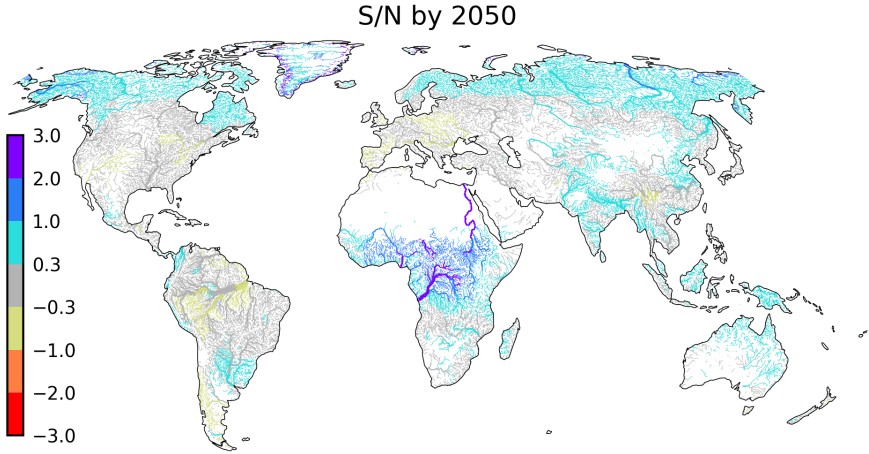

**Figure 6.** (a) Global map of signal-to-noise ratio of river flow by 2050. The ratio is the ensemble mean of the signal-to-noise ratio of each simulation. Rivers with little climatological flow ($< 5 \ \mathrm{m}^3\mathrm{s}^{-1}$) are masked out.

## 3.4 Regions susceptible to significant changes

The global evaluation of the river flow anomalies, the signal-to-noise ratio, and the time of emergence reveals that there are four regions projecting strong changes for the coming decades and present a general consensus among models: Central Africa, the Arctic, South Asia, and Patagonia. Here, we focus the analysis on those regions where, according to the projections, the signal of climate change of mean flow either emerges or closely approaches the threshold of its familiar climatology (see Fig. 8).

The most severe changes are projected for rivers originating in Central Africa. Several important rivers, such as Congo (which contributes the second largest discharge of all rivers in the world), Nile (considered the longest river in the world), as well as Sanaga, Niger, Volta, and Lake Chad tributaries (with important populations living upstream) are projected to experience a strong rise of their mean flow for the coming decades. Projections indicate an extra average discharge ranging from $\sim$16 % for the Sanaga River to around $\sim$49 % for the Lake Chad tributaries (see map on Fig. 8a). The aggregated discharge anomaly of these rivers exhibits a steady evolution during the past century that changes to a positive trend at the beginning of the current century (see time-series on Fig. 8a). The ensemble mean simulated river flow rise of this group of rivers exceeds the upper threshold of the familiar climate by the year 2018, and it drifts from unfamiliar to unusual climate by 2037. While the ensemble dispersion suggests a clear consensus in surpassing the familiar threshold, some models remain in the unfamiliar climate, others shift to an unusual climate, and some models progress to an unknown climate ($S > 3N$). This is closely related to the degree of global warming assumed by each GCM, with those projecting higher warming exhibiting a more pronounced signal of change.

The Arctic Ocean plays two roles in the global ocean circulation: it provides a pathway to connect the Pacific and the Atlantic oceans, and it receives Atlantic inflow, cools the water, and returns it to the Atlantic (Rudels and Friedrich, 2000). According to the simulations, this circulation could be affected in the near future. The projections indicate an average extra freshwater inflow

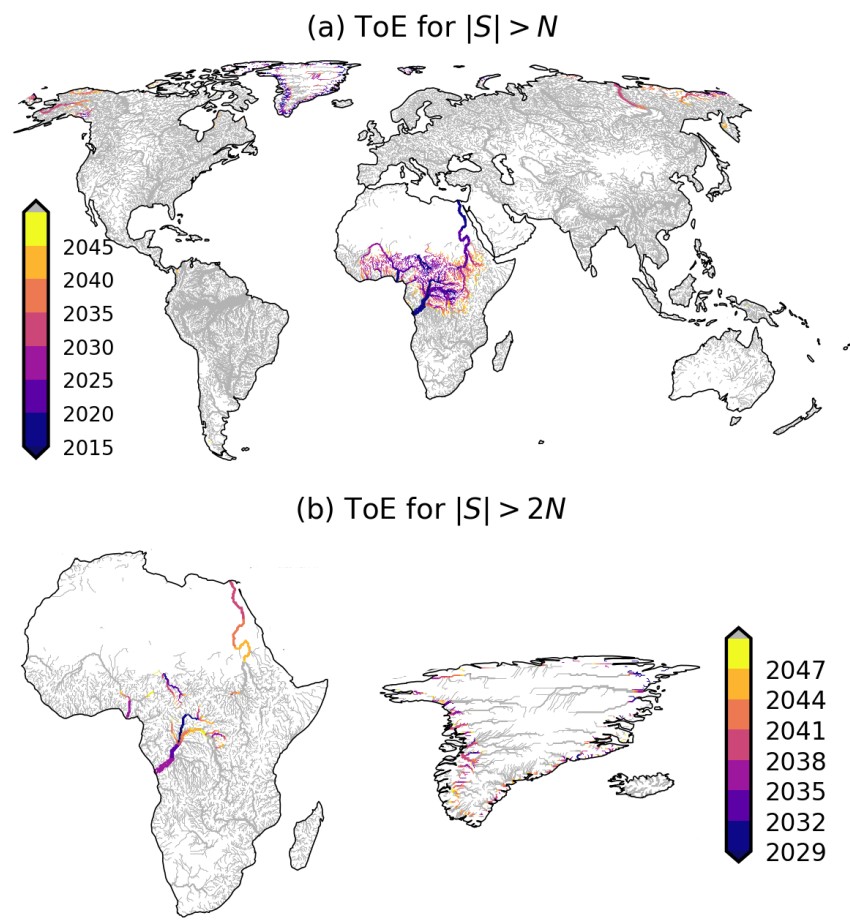

**Figure 7.** (a) Global map of ToE for river flow signal >1 or <-1. (b) Africa and Greenland maps of ToE for river flow signal >2 or <-2. Rivers that remain within the range of natural variability $[-N, N]$ until the end of the simulation are shown in grey. Rivers with little climatological flow ($< 5\,\mathrm{m}^3\mathrm{s}^{-1}$) are masked out.

of ∼12 % to the Arctic Ocean, which means that it will experience a freshening of its waters. The main contributors to this freshening are Yukon and Mackensie in North America, Greenland rivers, Ob, Yenisey, Lena, Indigirka, Kolyma, and Anadyr in Russia, which exhibit positive anomalies ranging from ∼6 % to ∼43 % (see map on Fig. 8b). The evolution of the integrated discharge anomaly of these rivers shows a rise of the signal after ∼2000, emerging to an unfamiliar condition by 2039, from then on stabilizing the signal slightly above the upper natural variability threshold. The ensemble dispersion presents a small spread around the ensemble mean signal, with some models remaining within the natural variability range until 2050.

The South Asia region also exhibits noteworthy anomalies (see left panel on Fig. 8c), and while these anomalies are projected to stay within the bounds of natural variability (see Figs. 6 and 7), we specifically emphasize this region due to its status as the

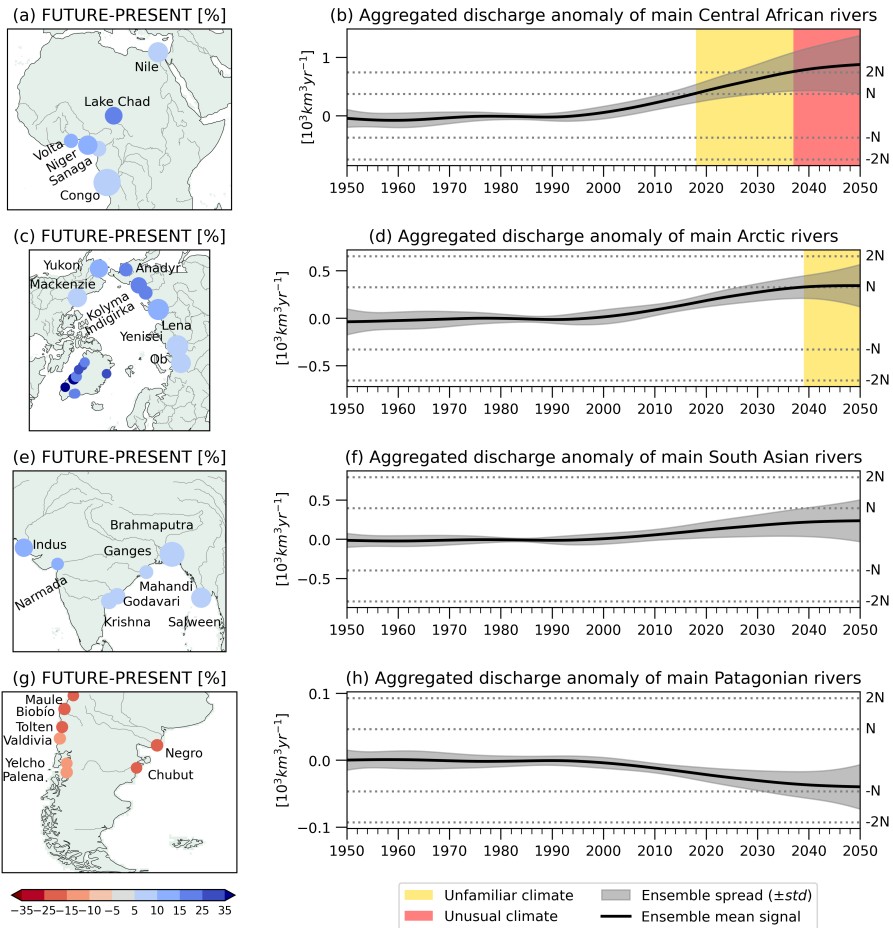

**Figure 8.** Regions displaying a consensus among models concerning their projected anomalous trends: (a) Central Africa, (c) the Arctic, (e) South Asia, and (g) Patagonia. The left panels illustrate the percentage difference in mean flow at the river mouth between FUTURE and PRESENT. The corresponding right panels showcase the aggregated discharge anomaly signal for the ensemble mean (along with the spread across models) for the rivers presented in the left panels.

most densely populated area globally. The significance lies in the potential impact of these anomalies on millions of people. The main rivers on the area of interest are Indus, which projects anomalies of ~28 %, and Narmada, Krishna, Godavari, Mahandi, Ganges-Brahmaputra, and Salween, which project anomalies of around 8 % (see left panel of Fig. 8b). Similar to the previous regions, and to what is observed globally, there are no significant changes in the signal until the early years of the present century, when a positive trend begins and extends until 2050 (right panel of Fig. 8b). The signal dispersion indicates that most models remain within the familiar climate, with some exceptions emerging to unfamiliar conditions by ~2040.

Although positive river flow trends dominate the projections, there are few regions of the world where models project drier conditions for the future. From these regions, Patagonia in South America stands out for the strong agreement among models

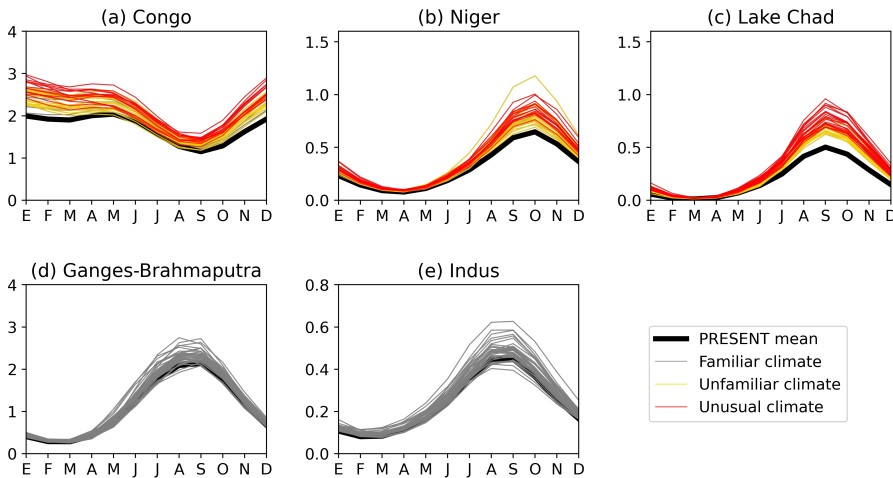

**Figure 9.** Annual cycles of (a) Congo, (b) Niger, (c) Lake Chad tributaries, (d) Ganges-Brahmaputra, (e) Indus at their mouths. Units are in $[10^3 km^3 yr^{-1}]$.

(see Figure 3). In particular, the models predict that the river flow of rivers originating in southern Andes and discharging into the Pacific Ocean (Maule, Biobío, Tolten, Valdivia, Yelcho, and Panela) or in the Atlantic Ocean (Negro and Chubut) will decline in a range that varies from 10 % to 18 % (see map on Fig. 8d). The signal time-series shown in Fig. 8d (right panel) shows that these flow reductions are not strong enough to emerge from the natural variability range before 2050, but this is likely to occur in the second half of the century if its signal strengthens in the long-term ($S/N = -0.9$ in 2050). Moreover, the signal spread indicates that some models shift to unfamiliar drier conditions after 2030. Patagonian rivers are snow-fed rivers that extend over arid to semi-arid regions. The river flow decline results from a combination of significantly reduced precipitation with increased evapotranspiration (see Fig. C1), which produces snow loss in the headwaters (Hoerling et al., 2019; Milly and Dunne, 2020).

Lastly, the regional assessment is complemented with a brief description of specific rivers at their mouths projecting significant changes in their annual cycle, which could provide useful insights for planning purposes given the population living upstream. Figure 9a-c show the cases of Congo, Niger, and Lake Chad tributaries in Central Africa, which present two common features. First, all annual cycles in the FUTURE period (2015-2050) are above the mean annual cycle of the PRESENT period (1950-2014). Second, these differences are amplified during their corresponding peaks, which could double for Niger and Lake Chad, or be up to ~50 % higher than the climatological peaks for the case of Congo. Focusing on Ganges-Brahmaputra and Indus rivers (9d-e) the results show exactly the same pattern, albeit slightly weaker. In these cases, the amplitude of most annual cycles will be exacerbated up to ~40 %, due to a strong increase of flow during the wet season.

## 4 Discussion and conclusions

We conducted 18 high-resolution global hydrological simulations for the period 1950-2050, forced by runoff from a variety of HighResMIP-CMIP6 GCMs, aiming to examine the effects of global warming on river flows worldwide in the coming decades. The comprehensive validation across global catchments indicates a consistent and commendable performance of our simulations for the PRESENT period (1950-2014), with particular skill observed for the 18-model ensemble mean.

The assessment of hydrological simulations involved estimating the S/N ratio and ToE to determine, for which rivers of the world, and when, the climate change signal will emerge from the natural variability. The method used for the calculation of the signal (both global and local) is key, as it determines the noise terms and the ToE. On one hand, the selection of filter, window function, and window length is key for estimating the global signal. Guided by rigorous sensitivity tests of various options, our analysis favored the use of a Lowess filter with a Hanning window of 41-year length. Our results indicated that the choice of filter significantly impacts the signal, as some filters introduce phase distortion and/or a "blending" effect, leading to erroneous high noise values that alter the ToE. These drawbacks are minimized by the Lowess filter, which effectively emphasizes long-term variations without introducing any distortion. Despite the subjectivity inherent in these choices, our analysis revealed that the resulting noise term exhibited low sensitivity to reasonable options of window function and length. A Hanning window of 41-year length was chosen as it effectively emphasizes long-term variations, essential for capturing climate change signals amidst natural variability. On the other hand, local signals can be estimated through linear regression of the global signal or by filtering local time-series. Both alternatives were tested for rivers with different regional trends (positive, neutral, and negative). The results revealed that the local filtered signals are affected by the filter's "blending" effects on extremes, regardless of regional trends. However, this issue was effectively mitigated when linear regressions were applied. These methodological considerations enhance the reliability of our S/N ratio estimation for assessing ToE.

Consistent with the IPCC AR6 findings (Douville et al., 2021; Caretta et al., 2022), the GCMs project a global intensification of land water budget components in the FUTURE period, showing increased precipitation (+3 %), evapotranspiration (+2 %), and runoff (+6 %) across simulations and resolutions. The enhanced runoff results in a clear positive global river discharge trend from ∼2000 in our simulations. The signal of the ensemble mean emerges to unfamiliar climate conditions by 2017 and to unusual climate conditions by 2033. Nonetheless, there is a large spread in the magnitude of the climate change signal among models, with global anomalies ranging from almost no change to +19 % by 2050. It can be argued that the ensemble mean climate change signal is strongly influenced by the high-resolution versions of GCMs, which simulate greater anomalies, and thereby, sooner ToE. This is particularly true for HadGEM3-GC31, the GCM with the highest anomalies in Africa (not shown). However, Müller et al. (2021b) provide robust evidence that high-resolution GCMs notably enhance the representation of land-atmosphere interactions in Africa through improved large-scale circulation and better-resolved local processes. Moreover, Müller et al. (2021a) showed that high-resolution HadGEM3-GC31 simulations notably improves the performance in mountainous regions, due to the finer definition of the orography, which favours the development of orographic precipitation and more runoff, matching better with river flow observations. While uncertainty in future projections is unavoidable, these

studies suggest that the GCMs projecting stronger changes for the near future are, nonetheless, those that demonstrated high skills in simulating complex processes in crucial regions.

Central Africa emerges as a focal point for notable hydrological changes, projecting significant increases in river flows that demand special attention in the near future. The ensemble mean signals for aggregated rivers, surpass familiar climate thresholds by 2018, transitioning to unusual conditions by 2037. Moreover, most projected annual cycles are above the mean annual cycle of the base period, but, most importantly, the amplitude of the cycles is intensified, being the annual highs the months with largest differences with respect to the historical values, with some peaks doubling in some cases. Such large changes may produce severe floods, with catastrophic consequences, given the vulnerability of the region. Indeed, the Congo River has suffered frequent floods recently (e.g., the severe floods from October 2019 to January 2020 reported in UNOCHA 2021) affecting at least 100,000 people per year since 2015 in Republic of Congo and Democratic Republic of Congo (Ritchie and Roser, 2014). Nigeria suffered an unprecedented flood in 2012 with 7,000,000 people affected and 363 reported deaths (Amangabara and Obenade, 2015), but also flooding events in 2018, 2020 (Ritchie and Roser, 2014), and 2022. IFRC (2022) reported at least 2,800,0000 people affected and at least 603 lives lost in the 2022 flood, being the near delta states the most affected. Similarly, Logone and Chari overflowed their banks, hitting N'Djamena in 2012 and 2022 (UNITAR, 2012; UNOCHA, 2022; Ritchie and Roser, 2014). However, the positive trend of Lake Chad tributaries is not necessarily bad news for the surrounding region. Lake Chad, a vital resource providing food and water to 50,000,000 people, has lost 90 % of its area since the 1970s (Gao et al., 2011). In agreement with the simulations, recent observation-based research has reported a recovery of the lake surface water extent and volume since 2000's (Pham-Duc et al., 2020), which brings hope to the surrounding growing communities.

The Arctic region stands out as a notable hotspot for hydrological changes, with simulations agreeing in projecting an average extra freshwater inflow of approximately 12 % into the Arctic Ocean. Major contributing rivers for these anomalies include Yukon, Mackenzie, Greenland rivers and the largest Russian rivers. The integrated discharge anomaly for these rivers indicates a discernible rise post-2000, shifting to unfamiliar conditions by 2039. The predicted extra discharge may influence a wide range of physical, chemical, and biological systems (Mankoff et al. 2020 and references therein). For instance, the enhancement of freshwater release may produce a freshening of the Arctic Ocean (Morison et al., 2012), which in turn affects the ocean stratification, the sea ice formation or melt, and potentially the global ocean overturning circulation (Solomon et al., 2021). Observation-based research aligns with our model simulations, reporting an accelerated rise of Arctic freshwater input from rivers since the 90's, favouring the cooling and freshening process (Rabe et al., 2011; Perner et al., 2019; Shiklomanov et al., 2021).

In South Asia, our models consistently project positive anomalies for major rivers such as the Indus and Ganges-Brahmaputra. While these anomalies remain within the bounds of familiar climate, a distinct positive trend has been simulated since around 2000, persisting until the end of the simulated period. This trend holds particular significance for the region, given the dense populations residing in the catchment areas. Floods in South Asia present a significant and recurring challenge, due to the prevalent monsoon climate, where approximately 80 % of the rainfall occurs during the wet season, resulting in a distinct annual river flow cycle characterized by prominent peaks. The projected positive trend in the near future may exacerbate the

frequency of floods, worsening the existing problem. Indeed, various parts of India have witnessed devastating flooding events in recent years (Hunt and Menon 2020 and references therein), leading to economic damages and hundreds of fatalities.

Most regions in the world present positive flow trends. A clear exception is Patagonia, whose rivers projecting flows that will remain in the range of natural variability but with a decline of about 15%. Patagonian rivers are snow-fed rivers that extend over arid to semi-arid regions, where water is used for electricity generation, agriculture, and consumption. Thus, the hydrological deficit of these rivers reduces the hydropower generation due to low dams levels, but also reduces the availability of water for irrigation affecting agriculture and livestock. In agreement with our results Rivera et al. (2021) reported frequent hydrological

droughts in the last decade, due to the reduced snow accumulation over the Andes, negatively impacting communities depending on these rivers. For instance, the continuous low levels of the river systems motivated Argentinian authorities to declare the water emergency for the catchments of Limay, Neuquén, and Negro in 2022, limiting the operation of the dams to guarantee the availability of water in the affected areas.

Recent studies reveal significant advancements in understanding the complex interplay between climate change and river

flow projections. Bosmans et al. (2022) introduced a high-resolution dataset projecting global river flow under various climate scenarios, which resembles the patterns observed in our findings. Zhou et al. (2023), in agreement with our projections, attribute changes in runoff to shifts in land surface characteristics such as vegetation and soil conditions. Zhang et al. (2023) predict similar river flow anomaly patterns to those found in our work but suggest that global river flow may be lower than projected by GCMs, attributing this discrepancy to the heightened sensitivity of river flow to changes in evapotranspiration, linked to

the phenomenon of vegetation greening. Our work complements these studies by employing advanced techniques such as the S/N ratio and the ToE, which are key for identifying when river systems may exhibit conditions beyond their known historical variability. Our findings underscore the pressing need for a paradigm shift in prioritizing water-related concerns in the context of climate change, as emphasized by Douville et al. (2022). Moreover, our study emphasizes the interplay between water cycle alterations and potential hydrological impacts, providing valuable insights for planning purposes. It is concerning that several

major rivers are projected to imminently surpass the bounds of their natural variability. However, the hydrological predictions presented in this work should be interpreted in the context of a very high baseline emission scenario, i.e., an outcome only likely if society does not make concerted efforts to reduce greenhouse gas emissions (Van Vuuren et al., 2011). In future work, we will extend the analysis to encompass a broader set of the new SSP scenarios.

## Appendix A: Internal variability in projections of runoff anomaly

Deser et al. (2012) identified a pronounced sensitivity of precipitation projections compared to temperature in their analysis of 40 members in regional climate model simulations for North America. In our experiments, where each GCM is represented by a single ensemble member per resolution and simulation type, this poses a challenge as the river flow projections may be significantly influenced by internal variability, especially knowing the direct impact of precipitation on runoff. To comprehensively examine this challenge, we conducted an analysis on the internal variability of the global river flow projections of a total

of 58 individual realizations spanning various GCMs. The breakdown of realizations per GCM is provided in Table A1. The

realizations resulted from varying the initial conditions with a random perturbation to the initial conditions, offering a thorough exploration of the models' response. Note that GCMs from the MRI and NICAM families provide only one realization, limiting the extension of the analysis to these specific families.

Figure A1 exhibits the global runoff anomaly projections of each GCM along with shaded bands representing the internal variability of (a) CNRM-CM6-1, (b) EC-Earth3P, and (c) HadGEM3-GC31. The GCMs' families present consistent and robust results. While the internal variability tend to rise over time, it is smaller than the inter-model variability and comparable to the variability given by the GCMs' resolution. Consequently, it is asserted that our set of simulations is adequate for the proposed objectives and that more realizations are unlikely to substantially alter the presented results.

**Table A1.** Number of realizations per GCM for analyzing internal variability.

| GCM | AMIP | COUPLED |
| --- | --- | --- |
| CNRM-CM6-1 | 10 | 3 |
| CNRM-CM6-1-HR | 10 | 3 |
| EC-Earth3P | 3 | 3 |
| EC-Earth3P-HR | 2 | 3 |
| HadGEM-GC31-L* | 3 | 3 |
| HadGEM-GC31-MM | 3 | 3 |
| HadGEM-GC31-HM | 3 | 2 |
| MRI-AGCM3-2-H | 1 | — |
| MRI-AGCM3-2-S | 1 | — |
| NICAM16-7S | 1 | — |
| NICAM16-8S | 1 | — |
| TOTAL | 38 | 20 |

*=M for AMIP (HadGEM-GC31-LM) and

*=L for COUPLED (HadGEM-GC31-LL)

## Appendix B: Notes about the calculation of the S/N ratio

Our estimation of the S/N ratio follows the method proposed by Hawkins and Sutton (2012), with a thorough parameter selection process that includes sensitivity tests, which guided the rationale for our choices. The key sensitivity test includes the evaluation of different filters, window types, and window lenghts (as exemplified in Fig. B1). The tested filters are Butterworth, Chebyshev, Elliptic, FFT low-pass, and Lowess. The window types are Rectangular, Hamming, Bartlett, Blackman, and Hanning. The window lengths are 21, 31, 41, 51, and 61 years. The key findings are summarized as follows:

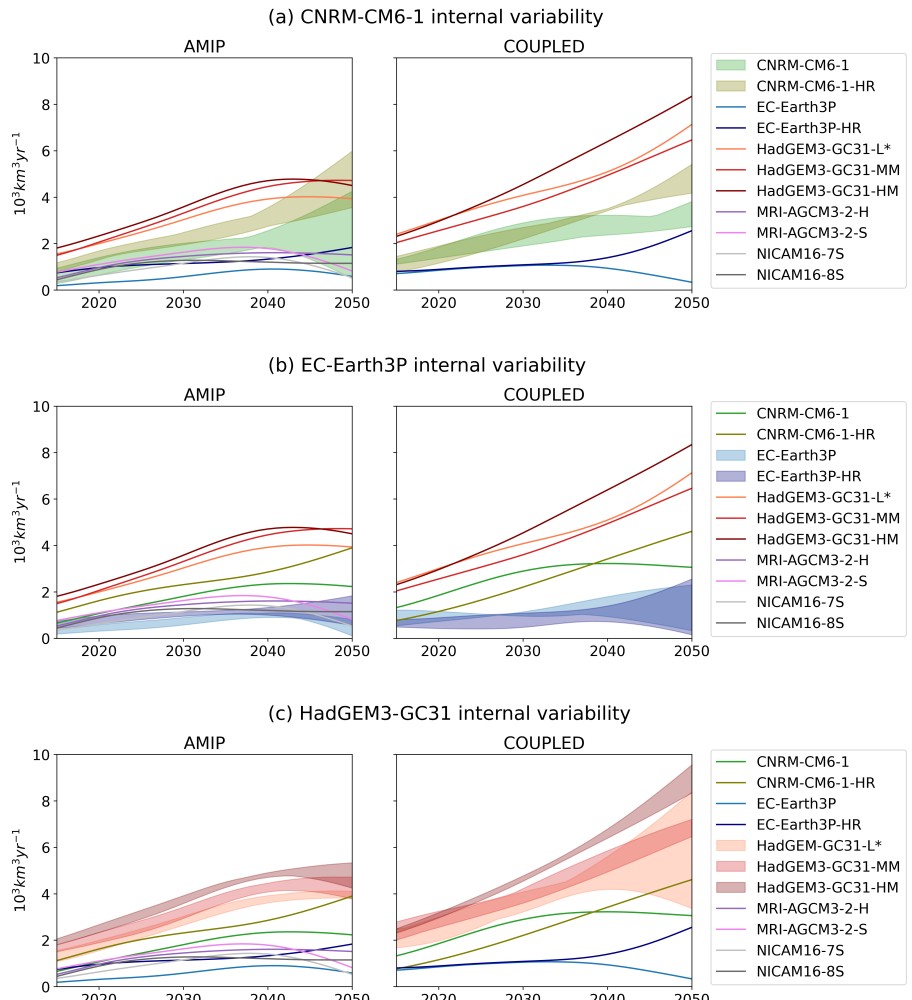

**Figure A1.** Low-pass filtered projections of global runoff anomalies for AMIP and COUPLED simulations of (a) CNRM-CM6-1, (b) EC-Earth3P, and (c) HadGEM3-GC31. The solid lines show the projections of individual GCMs. The shaded bands show the internal variability of the GCM realizations at different resolutions.

*Choice of filter*: The choice of the low-pass filter is a critical step in our analysis, as it directly impacts on the noise term, and subsequently, the ToE. In our approach, the climate change signal is estimated as $S_G = Q_G * w$, i.e., convolving the global river discharge anomaly time-series $Q_G$ with a Hanning window $w$ that has a length of 41 years. This operation performs a smoothing of the given time-series. The filter is chosen considering the following factors: (a) it effectively attenuates high-frequency noise without introducing phase distortion, unlike Butterworth, Chebyshev, or Elliptic, and minimizes boundary effects, unlike the FFT low-pass filter (see Fig. B1a); (b) the Hanning window provides similar smoothing results to other filters, except for the rectangular window, which introduces some spurious interannual variability (see Fig. B1b); and (c) a window of 41 years highlights the long-term variations (see Fig. B1c). Filters that produce phase distortion can exaggerate

differences between the original and filtered time-series, leading to a misrepresentation of the noise term with a higher value. For example, N ranges from 1.18 to 1.23 for filters that introduce phase distortion, whereas N=1.04 for the Lowess filter (Fig. B1a). Similarly, the "blending" of the FFT smoothed signal on the edges of the time-series may make the signal to unrealistically emerge or immerse on the natural variability range. The type and length of the window have a relatively minor impact in comparison. In summary, while we recognize that the selection of window length and smoothing options involves some subjectivity, the resulting $N$ term exhibits relatively low sensitivity to reasonable choices, varying less than 9 % among the entire set of window types and lengths tested for the global signal (Fig. B1b-c). Moreover, tests performed on local cases (not shown) demonstrates the insensitivity of the local signal and noise to different filtering options tested for the global signal.

*Linear regression*: While it may be argued that local signals cannot be effectively regressed with respect to the global signal, given that river flow anomalies are not homogeneous worldwide, we justify this approach for several reasons. Firstly, linear regression has been successfully applied to precipitation, which exhibits stronger heterogeneity than river flow, as seen in Hawkins et al. (2020) and IPCC AR6 Ch4 WGII Caretta et al. (2022). Secondly, river flow anomalies, though not entirely homogeneous, present consistent spatial responses at the catchment scale. Lastly, alternative methods, such as estimating local signals by filtering local time-series, can produce misleading results. For instance, we tested this alternative for three rivers with different regional trends in their future mean flows: Congo (positive), Amazon (neutral), and Negro (negative). The results reveal that local filtered signal is affected by the filter's "blending" effects on extremes, regardless of regional trends. Moreover, this "blending" yields to high S/N ratio values in the first year of simulation, which is unrealistic. This issue is avoided when linear regressions are applied.

*Natural variability*: In our study, we adhere to the definition of "natural variability" as proposed by Hawkins and Sutton (2012), also applied in Hawkins et al. (2020), i.e. noise is the local component that is unexplained by long-term global changes. We acknowledge that in some studies, the definition of noise has been broadened to encompass the uncertainty in the climate response to anthropogenic forcing and the uncertainty in future anthropogenic emissions (Giorgi and Bi, 2009; Hawkins and Sutton, 2009, 2011). However, the GCM simulations used in our study do not include anthropogenic alterations of the local to regional hydrological systems. Thus, our primary focus remains on the natural internal variability of climate, as this serves as the key source of noise relevant for the analysis of the ToE. The base period for the anomalies calculation covers 6.5 decades (1950-2014) capturing different variability time-scales from interannual to multidecadal. In summary, while we recognize that river flow variations in the "real world" include both natural and human-induced elements, our study primarily aims to assess when simulated changes in river discharge become distinguishable from the background of this natural climate variability.

In conclusion, the careful analysis underpins our methodological approach. These considerations ensure the robustness of the estimation of the S/N ratio in assessing the ToE.

## Appendix C: Changes in land precipitation, evapotranspiration, and surface temperature

The IPCC AR6 reports a strengthening of the future global land water budget components with strong regional variations (Douville et al., 2021; Caretta et al., 2022). The results presented in section 3.2 agree with the previous statement at the global

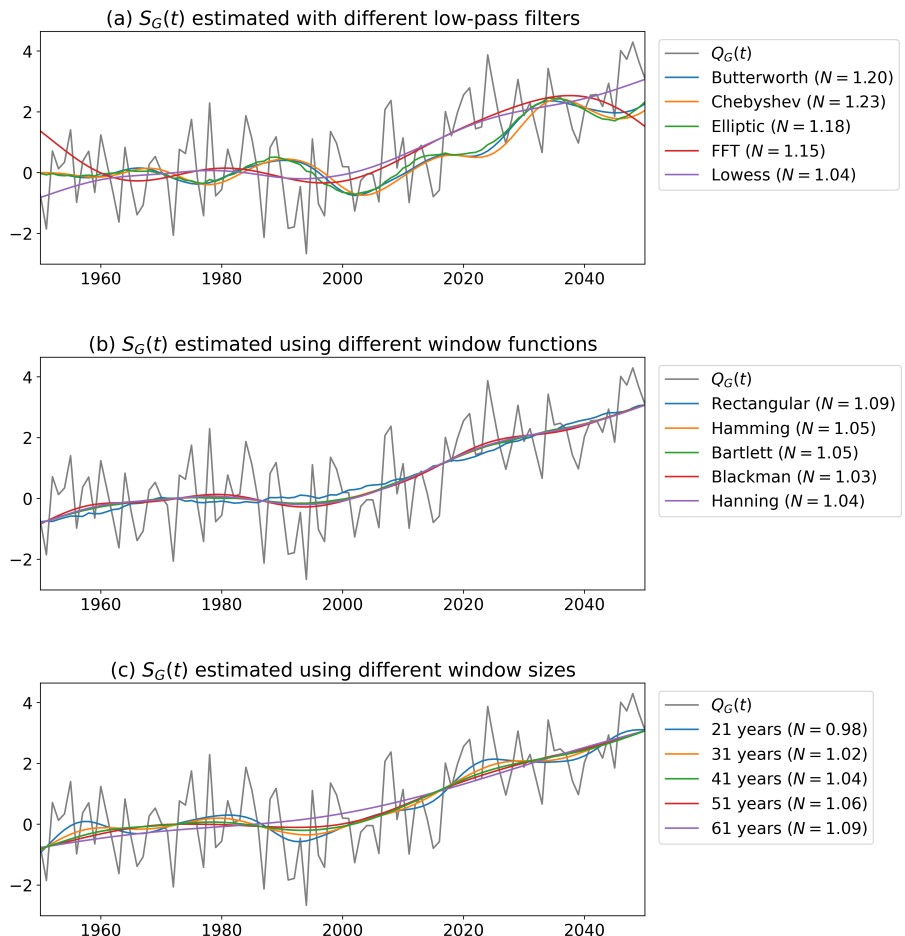

**Figure B1.** Low-pass filter selection test for the global river discharge anomaly time-series ($Q_G$) using the CNRM-CM6-1-HR model as a case study. The signal $S_G$ is estimated with various (a) filters, (b) window functions, and (c) window sizes. The (b) and (c) tests are applied for the Lowess filter. The resulting noise value of each filter is provided in the legend.

scale. Here, we complement those results with focus on the spatial variability of the projected changes in the water cycle. Figure C1a-b complements the Fig. 3 with the maps of the ensemble mean difference between FUTURE and PRESENT for land precipitation and evapotranspiration. These maps present similarities with the runoff map, mainly in the positive changes in the northern high latitudes, the Maritime Continent, and over the Sahel, which dominate the overall intensification of the water cycle.

The wetter conditions in the northern high-latitudes observed in Figs. C1a-b are associated with the well-known polar amplification of warming observed in Fig. C1c, however the specific processes responsible for this connection are still a topic of discussion. Some studies explain this relationship in terms of the moisture budget, arguing that either the increased surface evaporation following the retreat of sea ice and glaciers and the thawing of permafrost (Bintanja and Selten, 2014; Kopec et al.,

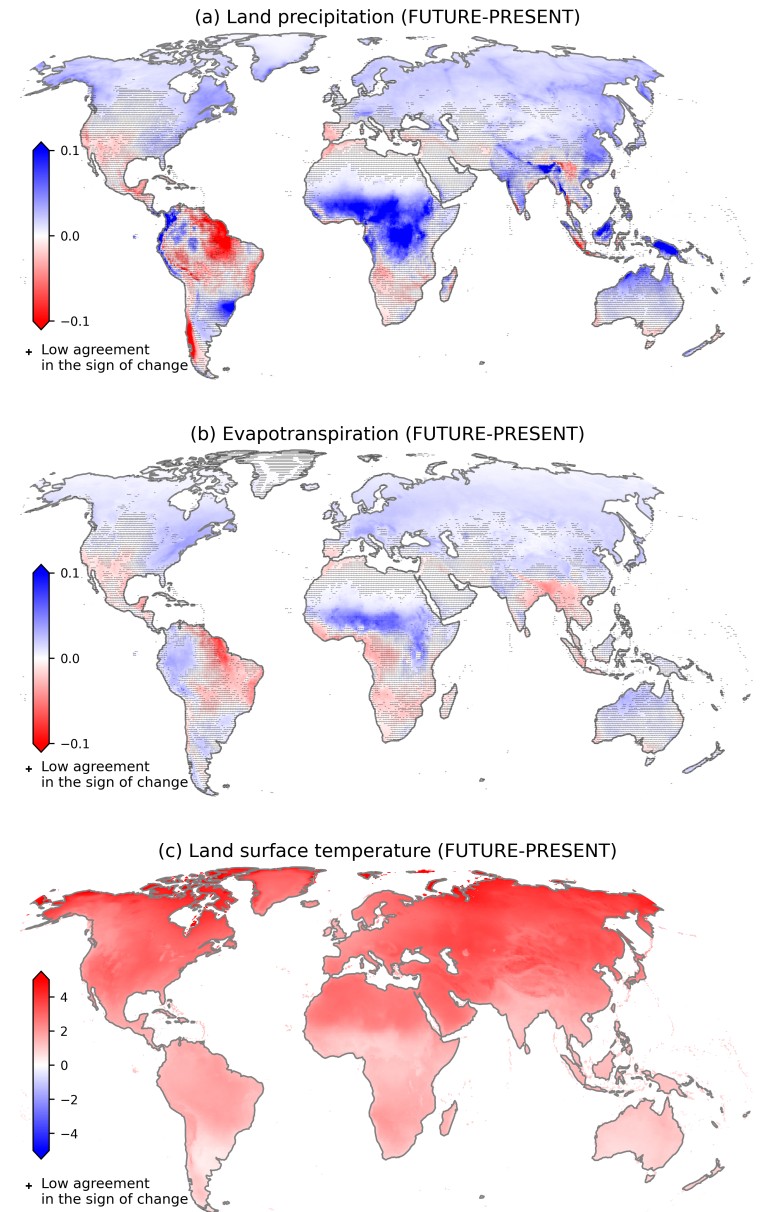

**Figure C1.** Multi-model ensemble mean differences in (a) land precipitation $[10^3\text{km}^3\text{yr}^{-1}]$, (b) evapotranspiration $[10^3\text{km}^3\text{yr}^{-1}]$, and (c) surface temperature [C] between FUTURE (2015-2050) and PRESENT (1950-2014). The crosses indicate that at least 3 out of 18 GCMs disagree in the sign of change.

470 2016) or the stronger moisture advection from lower latitudes (Bengtsson et al., 2011) cause increased precipitation, while Pithan and Jung (2021) support that it is mostly driven by stronger radiative loss of energy to space. Regardless, the rise of

precipitation and runoff alter the hydrological dynamic of the rivers flowing in cold regions (Barnett et al., 2005; Wang et al., 2021).

The wettening in intertropical regions is associated with the strengthening of the ITCZ. According to previous studies, the ITCZ presents a drying tendency at its edges but a strong moistening tendency in its core (Lau and Kim, 2015; Byrne et al., 2018; Douville et al., 2021). This is attributed to the intensification of ascending motion over the equatorial tropics, which elevates cloud tops, promotes convection processes, and leads to increased intense precipitation (Su et al., 2017). The increased land precipitation is partitioned in extra evapotranspiration and runoff in the Maritime Continent and over the Sahel (Figs. C1a-b and Fig. 2). However, there are regions presenting some different features. For instance, the Congo basin projects a strong rise of precipitation but combined with a slight evapotranspiration decrease, which favour the strong rise of runoff. This makes it the region of the world with the largest increase in river flow.

Lastly, there are other regions of the world projecting drier conditions of the hydrological cycle that partially compensate its global intensification. For instance, northern Brazil that exhibits reduced precipitation, evapotranspiration, and runoff, or Southern Andes, which shows a strong decay in precipitation combined with enhanced evapotranspiration, which deepens the decrease of runoff and river flow for the rivers that originate there (see Figs. C1a-b and Fig. 2).

*Code and data availability.* The HighResMIP CMIP6 GCMs simulations that provide the forcings are freely available on the Earth System Grid Federation (ESGF; esgf-node.llnl.govprojects/cmip6). The TRIPpy model used in this research is freely available on its GitHub repository (https://github.com/ovmuller/TRIPpy, Müller 2023). The repository provides access to the source code, documentation, and examples for running simulations at both global and regional scales.

The simulations are stored in JASMIN, the U.K. supercomputer for environmental science deployed on behalf of the Natural Environment Research Council (NERC). They are accessible upon request to the corresponding author.

*Author contributions.* OVM conceived the research question, performed the simulations, computed and interpreted the results, and wrote the first draft of the paper. PCM, PLV contributed to the conceptualization of the research and the interpretation of the results. EH analyzed and assured the quality of the results given his expertise in ToE methods. All the authors revised the paper and agreed to the submitted version.

*Competing interests.* The authors declare that they have no conflict of interest.

*Acknowledgements.* We specially thank to Chihiro Kodama from JAMSTEC and Bertrand Decharme from CNRM for useful comments and references. This research was supported by the PRIMAVERA project funded by the European Union's Horizon 2020 Research and Innovation Programme under Grant Agreement 641727. OVM acknowledges further support from the grants PEICID-2021-028 from the ASaCTeI, PICT-2019-2019-03982 from the ANPCYT, and PIP 11220200102257CO from the CONICET.

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
