# Peer review of "River flow in the near future: a global perspective in the context of a high-emission climate change scenario"

_EGUsphere, 2023_

## Author Comment (AC1)

**Responses to Referee #1**

We appreciate the detailed and constructive feedback provided by the anonymous reviewer, which has been valuable in enhancing the manuscript. We respond by highlighting the reviewer's comment in shaded text and our responses in black.

**General comments:**

The aim of this work is to provide a global assessment of near future river flow changes within the context of a high-emission climate change scenario. The authors leverage the TRIP model to simulate river flow dynamics, which is forced by runoff from HighResMIP simulations. They then quantify the changes by using the signal to noise ratio and time of emergence metrics. I have two major concerns on this work: 1) the need for a thorough justification, validation, and demonstration of signal and noise estimations; 2) the absence of logistical and substantive content that elucidates the implications for metropolitan areas and water resource management. Please see my detailed comments below.

We appreciate the reviewer's detailed feedback and their two major concerns regarding our manuscript. The comments provide valuable insights into the clarity and robustness of our work. First and foremost, we would like to address the following:

1) We understand the importance of providing a strong rationale for our signal and noise estimations. To address this concern, we provide a comprehensive and detailed description of our signal-to-noise and time of emergence methodology in the responses to the main concerns, and explain the main points of that in the revised manuscript. This expanded explanation sheds light on the utility of these techniques for our research and includes specific validation tests, ensuring a thorough justification, validation, and demonstration of our estimations.

2) We recognize that it is challenging to establish cause-and-effect relationships between simulated climate change signals and particular extreme events and/or sectoral impacts. Following reviewer #2's suggestion, we will replace the section "Rivers susceptible to strong changes" by "Regions susceptible to strong changes". In this new section we analyze the discharge projections grouping rivers in four regions with (a) trends that significantly deviate from the historical period, and (b) clear consensus among the model simulations. The regions are: Central Africa, Arctic, South Asia, and Patagonia. This regional analysis provides a more concise way to convey the broader implications of our study. The information about the potential risks for individual rivers will be briefly summarized at the end of the new manuscript.

**Major comments:**

1. Abstract: the flow of the abstract from L9 to L14 is hard to follow. They did not conclusively reflect what has been discussed in the main text. Please consider refine the abstract.

The abstract will be rewritten to more accurately reflect the main findings of our research, in particular, those referring to the new analysis at regional scale (explained in major comment 7).

2. L77-79 and Appendix A: additional evidence is necessary to support the hidden assumption that all models share similar internal variability as HadGEM-GC31. Do the remaining models contain ensemble members of precipitation? If so, how does the precipitation spread across the ensemble members? Do they share the similar internal variability as the precipitation spread of HadGEM-GC31? Consider exploring such variables that could influence runoff to earn the statement.

We agree with the reviewer regarding the assumption of similar internal variability across all models. To address this concern, we obtained access to additional realizations of the HighResMIP-CMIP6 from the Laurence Livermore National Laboratory (LLNL) Earth System Grid Federation (ESGF) Node (esgf-node.llnl.gov), which includes total runoff among the variables. This extended our internal variability analysis to a total of 58 individual realizations across different GCMs.

The breakdown of realizations per GCM is provided in Table R1.1:

| GCM | AMIP realizations | COUPLED realizations |
|---|---|---|
| CNRM-CM6-1 | 10 | 3 |
| CNRM-CM6-1-HR | 10 | 3 |
| EC-Earth3P | 3 | 3 |
| EC-Earth3P-HR | 2 | 3 |
| HadGEM3-GC-31-L* | 3 | 3 |
| HadGEM3-GC-31-MM | 3 | 3 |
| HadGEM3-GC-31-HM | 3 | 2 |
| MRI-AGCM3-2-H | 1 | ---- |
| MRI-AGCM3-2-S | 1 | ---- |
| NICAM16-7S | 1 | ---- |
| NICAM16-8S | 1 | ---- |
| Total | 38 | 20 |

Table R1.1: Number of realizations per GCM used to analyze the GCM's internal variability.

Note that GCMs from the MRI and NICAM families provide only one realization, limiting the extension of the analysis to these specific model families.

The internal variability results for CNRM-CM6-1 and EC-Earth3P GCMs confirm our earlier findings for the HadGEM3-GC31 family. While the internal variability tends to rise over time, it is smaller than the inter-model variability and comparable to the variability given by the GCMs' resolution. These new results strengthen the robustness of our assumption that the set of simulations is adequate for the proposed objectives and that more realizations would not substantially alter the presented results.

New panels showing the internal variability of CNRM and EC-Earth3P GCMs' families (see Figure A1) along with their interpretation will be added to the Appendix A: "Internal variability in projections of runoff anomaly".

[Figure]

Figure A1. Low-pass filtered projections of global runoff anomalies for AMIP and COUPLED simulations of (a) CNRM-CM6-1, (b) EC-Earth3P, and (c) HadGEM3-GC31 simulations. The solid lines show the projections of individual GCMs. The shaded bands show the internal variability of the GCM realizations at different resolutions.

3. While the authors direct readers to Müller et al. (2021a) for details on the TRIP model, it would be beneficial to provide a brief overview of TRIP's global-scale performance and the rationale for its selection.

Thanks for making this important point. To address this concern:

(a) We will add more information about the model origin and features that make it ideal for our purposes in the section "2.1 GCM simulations and river routing model". The model's key attribute lies in its simplicity, enabling long-term global simulations with minimal computational resources, all while delivering commendable performance (as shown in item b).

(b) We will include an explicit validation of simulated river flows following the suggestion of reviewer #2. It includes the calculation of four different metrics: (relative bias, overlapping coefficient, correlation coefficient, and non-parametric Kling-Gupta efficiency) for the 18 hydrological simulations and the ensemble mean. The metrics are based on the comparison of our simulations against monthly observations of a selection of 346 monitored rivers, which cover about 42% of the global land and their flows contribute to about 45% of the global river discharge. The validation results clearly demonstrate the robust performance of the model in this study (please see them in our response to reviewer #2 main comment #1). Further details about the validation methodology will be included in the section "2.2 Assessment methodology", while the scores will be analyzed in the new section "3.1 Validation of the hydrological simulations".

(c) We will provide a link to the GitHub repository of the used model in the "Code and data availability" section. The repository includes access to the source code, documentation, and examples for running at the global and regional scales.

4. L110-114: The demonstration of the estimation of signal and noise ratio needs more details:

a) the mathematical details of the low-pass filter, justification for its choice, and an explanation of how the choice of low-pass filter might impact the noise term;

b) regarding the noise term, why this could be a representation of "natural variability"? Do the variation of the river flow in the PRESENT period include impact of human water activities? If so, how would this impact be separated from the "natural variability"?

c) why the signal of a local river flow anomaly can be linearly regressed on the global signal given large spatial heterogeneity in local river flow variability.

These terms need to be better demonstrated in the method section and the robustness of the estimation of signal to noise ratio need to be proved.

The estimation of the signal-to-noise ratio follows the method proposed by Hawkins and Sutton (2012), with a thorough parameter selection process that includes sensitivity tests. We appreciate the reviewer's feedback and would like to provide a clear response to each concern.

a) Choice of filter:

The choice of the low-pass filter is indeed a critical step in our analysis, as it directly impacts on the noise term $N$, and subsequently, the Time of Emergence. In our approach, the climate

change signal $S_G$ is estimated as $S_G = Q_G * w$, i.e., convolving the global river discharge anomaly time-series $Q_G$ with a Hanning window $w$ that has a length of 41 years. This operation performs a smoothing of the given time-series. The filter is chosen considering the following factors:

(i) it effectively attenuates high-frequency noise without introducing phase distortion, unlike Butterworth, Chebysev, or Elliptic (see their phase shifts around 2020 in Fig. R1.1a), and minimizes boundary effects, unlike the FFT low-pass filter (see the blending effect in the first and last decades in Fig. R1.1a);

(ii) the Hanning window provides similar smoothing results to other filters, except for the rectangular window, which introduces some interannual variability (see Fig. R1.1b); and

(iii) a window of 41 years highlights the long-term variations.

Filters that produce phase distortion can exaggerate differences between the original and filtered time-series, leading to a misrepresentation of the noise term with a higher value. Similarly, the "blending" of the FFT smoothed signal on the edges of the time-seres, may make the signal to unrealistically emerge or immerse on the natural variability range. The type and length of the window have a relatively minor impact in comparison. In summary, while we recognize that the selection of window length and smoothing options involves some subjectivity, the resulting N term exhibits relatively low sensitivity to reasonable choices. Moreover, Fig. R1.2 demonstrates the insensitivity of local signal and noise to different filtering options tested for the global signal.

[Figure]

Figure R1.1. Low-pass filter selection test for a river discharge anomaly time-series ($Q_G$) using the CNRM-CM6-1-HR model as a case study. The signal $S_G$ is estimated with various (a) filters, (b) window functions, and (c) window sizes. The test of parameters in (b) and (c) is applied for the Lowess filter. The resulting noise value of each filter is provided in the legend.

[Figure]

Figure R1.2. Illustration of the robustness of local signal $(S_L)$ and noise $(N_L)$ derived from various filter options applied to the global signal. The time-series represents the Yukon discharge at its river mouth.

**b) Natural variability:**

In our study, we adhere to the definition of "natural variability" as proposed by Hawkins and Sutton (2012), also applied in Hawkins et al. 2020, i.e. noise is the local component that is unexplained by long-term global changes. We acknowledge that in some studies, the definition of noise has been broadened to encompass the uncertainty in the climate response to anthropogenic forcing and the uncertainty in future anthropogenic emissions (Giorgi and Bi, 2009; Hawkins and Sutton, 2009, 2011). However, the GCM simulations used in our study do not include anthropogenic water activities. Thus, our primary focus remains on the natural internal variability of climate, as this serves as the key source of noise relevant for the analysis

of the Time of Emergence. The base period for the anomalies calculation covers 6.5 decades (1950-2014) capturing different variability time-scales from interannual to multidecadal. In summary, while we recognize that river flow variations in the 'real world' include both natural and human-induced elements, our study primarily aims to assess when simulated changes in river discharge become distinguishable from the background of this natural climate variability, following the approach outlined by Hawkins et al.

c) Linear regression:

We agree with the reviewer that river flow anomalies are not homogeneous worldwide. However, we justify the estimation of local signals based on linear regression of local anomalies with respect to global anomalies considering that:

(i) It has been applied for precipitation which is strongly heterogeneous, even more than river flow. See Hawkins et al. (2020) and the IPCC AR6 Ch4 WGII (Caretta et al. 2022).

(ii) River flow anomalies are not largely heterogeneous, indeed they present consistent spatial responses at the catchment scale.

(iii) The alternative method to estimate the signal, i.e. estimating local signals by filtering local time-series produces misleading results. The local filtered signal is impacted by the filter's 'blending' effects on the extremes. This is clearly shown in Fig. R1.3 for Congo, Amazon, and Negro, three rivers with different regional trends. Figure R1.4 shows how the blending yields high SNR values in the first year of simulation, which is unrealistic. This issue is avoided when linear regressions are applied.

Following our above responses to each individual concern of comment 4, we will summarize the main points of this discussion and include it in the revised manuscript. This helps ensure that the methodological details are robustly presented and that the reader can follow the logic behind our decisions.

Caretta, M.A., A. Mukherji, M. Arfanuzzaman, R.A. Betts, A. Gelfan, Y. Hirabayashi, T.K. Lissner, J. Liu, E. Lopez Gunn, R. Morgan, S. Mwanga, and S. Supratid, 2022: Water. In: *Climate Change 2022: Impacts, Adaptation and Vulnerability.* Contribution of Working Group II to the Sixth Assessment Report of the Intergovernmental Panel on Climate Change [H.-O. Pörtner, D.C. Roberts, M. Tignor, E.S. Poloczanska, K. Mintenbeck, A. Alegría, M. Craig, S. Langsdorf, S. Löschke, V. Möller, A. Okem, B. Rama (eds.)]. Cambridge University Press, Cambridge, UK and New York, NY, USA, pp. 551–712, doi:10.1017/9781009325844.006.

Giorgi, F., and X. Bi (2009), Time of emergence (TOE) of GHG-forced precipitation change hot-spots, Geophys. Res. Lett., 36, L06709, doi:10.1029/2009GL037593.

Hawkins, E., and R. Sutton (2009), The potential to narrow uncertainty in regional climate predictions, Bull. Am. Meteorol. Soc., 90, 1095–1107, doi:10.1175/2009BAMS2607.1.

Hawkins, E., and R. Sutton (2011), The potential to narrow uncertainty in projections of regional precipitation change, Clim. Dyn., 37, 407–418, doi:10.1007/s00382-010-0810-6.

Hawkins, E., and Sutton, R. (2012). Time of emergence of climate signals. Geophysical Research Letters, 39(1).

Hawkins, E., Frame, D., Harrington, L., Joshi, M., King, A., Rojas, M., and Sutton, R. (2020). Observed emergence of the climate change signal: from the familiar to the unknown. Geophysical Research Letters, 47(6), e2019GL086259.

[Figure]

Figure R1.3. Local signal $(S_L)$ estimated by filtering with the Lowess filter (blue) compared to the local signal based on the linear regression of local anomalies with respect to global anomalies (orange). The signals are estimated for time-series of rivers presenting different trends (positive, neutral, and negative): (a) Congo, (b) Amazon (), and (c) Negro river flow at their river mouth.

[Figure]

Figure R1.4. Resulting SNR for years 1950 (top), 2000 (middle), and 2050 (bottom) when local signals are estimated by filtering with the Lowess filter (left) compared to the SNR when local signal based on the linear regression of local anomalies with respect to global anomalies (right).

5. In Figure 2, the difference in runoff between the two periods in Greenland and central Australia is small while in Figure 3 (a) the percentage changes in river flow for these locations are outstanding. Is this discrepancy due to the use of the percent change as a measure, rather than absolute differences? I think it would be beneficial to explain the discrepancy.

The visual discrepancy between the small and large differences observed in Greenland and central Australia in our original Figures 2 and 3a has two reasons:

a) mathematical: differences expressed as percentage changes tend to be high on places where mean values are small, such as Greenland or Australia.

b) visual: Figure 3a shows uniform colors at the catchment scale, and such colors represent the percentage change at the river mouth, which integrates the runoff of the entire catchments. For instance, the strong values of Greenland catchments in Figure 3a are directly influenced by the high difference between FUTURE and PAST near the deltas shown in Figure 2, rather than the small differences observed upstream.

We clarify this in the revised manuscript.

6. About ToE: again, the method of signal and noise decomposition would largely determine the results of ToE. I wonder if the author could check the robustness of the ToE as well.

We agree with the reviewer, the ToE is sensitive to the signal and noise estimations. In our response to the major comment 4, we provide a comprehensive justification of the methodology we used to estimate the signal-to-noise ratio, demonstrating the robustness of our approach. We will include this clarification in the revised manuscript, addressing the sensitivity of ToE to signal-to-noise calculations.

7. Figure 7: the discussion on the results shown in Figure 7 is confusing. In my perspective, changes are described in the context of a simulation scenario so that they are not yet scalable to real world situation. However, the authors mentioned about the risk to metropolitan areas and the implications to infrastructure management. I think these discussions are relevant but somehow feel disconnected from the results of the simulation. To better connect, I think the authors should first demonstrate how the simulated river flow for PRESENT period compare to observations and form the discussion based on it. This might also help condense the content in the abstract.

We appreciate the reviewer's feedback on the discussion about simulated river flow projections and extreme events observed in the real world. Recognizing that this connection is too ambitious that connection, we opted to follow the suggestion of reviewer #2 and replace that analysis of specific cases with the assessment of groups of rivers in regions with similar trends. This modification streamlines our analysis by grouping rivers into regions with notable trends and clear consensus among model simulations—Central Africa, Arctic, South Asia, and Patagonia. This is a more concise way to better connect with the simulation results and convey the broader implications of this study. Please see our response to reviewer #2 main comment #2.

**Minor comments:**

1. L68: Clarification is needed regarding what "availability"

Surface and subsurface runoff is simulated by all GCMs, but these variables are not always stored. The text will be rephrased to "These HighResMIP GCMs were selected based on the availability of surface and subsurface runoff data."

2. L86-87: Please specify the resolution of the target grid.

Thanks for noting it was unclear. We will rephrase it: "The simulations are run globally (excluding Antarctica) using the nearest-neighbour option to regrid the runoff from the original GCM resolutions to the target grid at a common resolution of 0.25°."

L97-98: the term "expected changes in rivers" requires clarification. Rationale is needed to connect the three steps.

We will rephrase the term as "projected changes in rivers". As indicated in our response to major comment 7 we will provide a regional analysis of the projections that better connects with the simulated projections and the evaluation of signal-to-noise ratio and time-of-emergence.

---

## Author Comment (AC2)

**Responses to Referee #2**

We thank the constructive insights offered by the anonymous referee, which significantly contributes to the improvement of the manuscript. We respond by highlighting the reviewer's comment in shaded text and our responses in black.

**Main Comments:**

The manuscript titled "River flow in the near future: a global perspective in the context of a high-emission climate change scenario" investigates the potential effects of global warming on river flows worldwide from 2015-2050 with hydrological simulations. The study aims to provide insights into the potential changes in river flows and their broader socio-environmental consequences. Overall, the manuscript is well structured and clear. However, the following points raise my concerns.

1) There seems to be a lack of explicit validation of simulated river flows by observations from the same historical period. In general, it is important to ensure that the modeling framework accurately reproduces observed river flows in the historical period before trusting projections for the future.

Thanks for making this important point. In response, we will incorporate a comprehensive validation of simulated river flows in the revised manuscript. This validation will involve assessing four distinct metrics: relative bias (RB), overlapping coefficient (OC), correlation coefficient (r), and non-parametric Kling-Gupta efficiency (npKGE) for the 18 hydrological simulations and the ensemble mean. These metrics, which evaluate diverse aspects of the simulations, are computed by comparing our simulations with monthly observations of a selected set of 346 monitored rivers, covering approximately 42% of the global land and contributing to about 45% of the global river discharge.

The ensemble mean presents a relative bias of 7.6%, an overlap coefficient of 0.63, a correlation coefficient of 0.72, and an efficiency of 0.58. These scores improve to -1.8%, 0.62, 0.76, and 0.71 (respectively) when the assessment is restricted to the 20 largest monitored rivers (see new Figure 1). Another important finding of the validation is the consistency of the scores across models. Independent of the resolution and the type of simulation, the scores remain in a narrow range. The validation results compellingly prove the robust performance of the model in the context of this study.

Further details about the validation methodology will be included in the section "2.2 Assessment methodology", while the scores will be analyzed in the new section "3.1 Validation of the hydrological simulations".

[Figure]

Figure 1. Validation of the global hydrological simulations. (a) Monitored rivers with black dots indicating the observation sites for river flow and colors highlighting the catchment area that contributes to each monitored point. (b) Average relative bias (RB), overlapping coefficient (OC), correlation coefficient (r), and non-parametric Kling-Gupta Efficiency (npKGE) for each simulation and the ensemble mean. (c) As (b) but for the 20 largest catchments. The averaged OC, r, and npKGE are calculated using a weighted average, where the weight assigned to each river is proportional to its contribution to the total observed flow under evaluation.

2. section 3.2.1 is overly descriptive, providing detailed information about individual rivers, their importance to their respective regions, historical context, and model projections. While such detail is valuable to some extent, it may be more than necessary for the main message of the article. I recommend a more concise way to convey the broader implications of this study. For example, group rivers with similar trends and mention the specifics only when they significantly deviate from the general trend. The implications for human settlements, global water systems, and climate systems can be summarized in one final paragraph. In addition, explaining the "why" behind the trend or time of emergence can provide more insightful analysis and make the findings more compelling. Also, the discussion section reads repetitively, which seems to summarize the above finding again.

We value the reviewer feedback regarding the detailed nature of section 3.2.1. While recognizing the value of such details, we acknowledge the need for conciseness in conveying

the primary message of the article. In response, we will adopt the suggested approach of grouping rivers with similar trends. We identify four regions with notable trends and clear consensus among model simulation: Central Africa, Arctic, South Asia, and Patagonia (see new Figure 8). This approach offers a concise way to communicate the wider implications of our study. This regional assessment will be complemented with a brief description of specific rivers at their mouths projecting significant changes in their annual cycle, which could be useful for planning purposes given the population living upstream (see new Figure 9).

In response to the reviewer's concern about repetitive elements in the discussion section, we are committed to enhancing clarity and conciseness in our revised manuscript. The updated section will delve into various topics, including choices related to the signal-to-noise ratio technique and their potential impact on results (discussed in our response to reviewer #1 major comment #4), the internal variability of model projections, and the potential sectoral implications of projected changes at the regional and at the local scale. We believe these refinements will effectively address the raised concerns and elevate the overall clarity of our discussion.

[Figure]

Figure 8. Percentage difference in mean flow at the river mouth between FUTURE and PRESENT for rivers presenting consensus among models regarding their trends in (a) Central Africa, (c) Arctic, (e) South Asia, and (g) Patagonia. The right panels depict the aggregated discharge anomaly signal for the ensemble mean (and the spread across models) for the rivers shown in left panels.

[Figure]

Figure 9. Annual cycles of (a) Congo, (b) Niger, (c) Lake Chad tributaries, (d) Ganges-Brahmaputra, (e) Indus at their mouths. Units are in $[10^3 km^3 yr^{-1}]$.

3. For the information in Appendix A, it's a bit of a leap to conclude that "it may be assumed that our set of simulations is adequate for the proposed objectives and that more realizations would not present substantial alter the presented results" without information about the internal variability of other models. While this may be true for the HadGEM3-GC31 model, it may be premature to state this as a broader conclusion without data from other models to support it.

We agree with the reviewer, the internal variability is only shown for HadGEM-GC31 and that can not be extended to all models. We got access to more realizations of the HighResMIP-CMIP6 from the Laurence Livermore National Laboratory (LLNL) Earth System Grid Federation (ESGF) Node (esgf-node.llnl.gov), which includes total runoff among the variables. Thus, the internal variability analysis was extended to these new GCM simulations, resulting in an assessment of 58 individual realizations in total.

The new results for the CNRM-CM6-1 and EC-Earth3P GCMs family confirm our previous findings for the HadGEM3-GC31 family. While the internal variability tends to rise over time, it is smaller than the inter-model variability and comparable to the variability given by the GCMs' resolution (please see the results in our response to reviewer #1 major comment #2). These new results provide robustness to the assumption that our set of simulations is adequate for the proposed objectives and that more realizations would not substantially alter the presented results.

The new figure showing the internal variability of CNRM-CM6-1, EC-Earth3P, and HadGEM-GC31 GCMs' families along with their interpretation will be incorporated to the Appendix A: "Internal variability in projections of runoff anomaly".

**Minor comments:**

1) Figure 4, can you explain why most of the solid lines show a downward trend after 2045?

The "boundary" effects observed in the first and last decades of the several GCMs' signals are caused by the filtering process. The choice of the low-pass filter is a critical step in our analysis. We tested several filters and selected it considering: (a) an effective attenuation of high-frequency noise without introducing phase distortion, and (b) the minimization of the boundary effects. Figure R1.1a shows how the various filter's respond to the river discharge anomaly time-series of a given GCM. From the various options, we selected the Lowess filter in our approach given that it does not introduce phase distortion, unlike Butterworth, Chebysev, or Elliptic (see their phase shifts around 2020), and minimizes boundary effects, unlike the FFT low-pass filter (see the strong blending effect in the first and last decades). Thus, the selected filter ensures realistic climate change signals and noise terms.

A detailed explanation of the choices related to the filtering process and the overall signal-to-noise technique is provided in our response to reviewer #1 major comment #4. The revised manuscript will summarize the main aspects of these topics.

[Figure]

Figure R1.1a. Low-pass filter selection test for a river discharge anomaly time-series ($Q_G$) using the CNRM-CM6-1-HR model as a case study. The signal $S_G$ is estimated with various filters.

2) Figure 5, I'm curious if the order of calculating the ensemble mean and calculating the signal-to-noise ratio would have an impact, let's say calculating the signal-to-noise ratio of the ensemble mean of the simulation instead.

The order of calculation has an impact on the results. Both alternatives present similar patterns (see Figures 5 original and suggested). However, calculating the  signal-to-noise ratio (SNR) of the ensemble mean tends to magnify the SNR values. This effect arises from the ensemble mean time-series, which inherently exhibits minimal interannual variability due to the cancellation of individual model variabilities. The reduced variability results in a small noise value, thereby yielding high SNR values. For this reason, we consider our original calculation to be more realistic.

**S/N by 2050**

[Figure]

Figure 5 (original). Global map of signal-to-noise ratio of river flow by 2050. The ratio is the ensemble mean of the signal-to-noise ratio of each simulation. Rivers with little climatological flow are masked out.

**S/N by 2050**

[Figure]

Figure 5 (suggested). Global map of signal-to-noise ratio of river flow by 2050. The ratio is the signal-to-noise ratio of the ensemble mean. Rivers with little climatological flow are masked out.

**Editorial comment:**

1) Figure 1, there seem to be no squares in the three main plots.

Thanks for noting it. We originally plotted the COUPLED GCMs with squares, but then decided to show them with closed circles, to allow a fair visual comparison of the warming level with the AMIP GCMs. We will fix the legend in the revised manuscript.

---

## Author Response (AR2)

**Responses to Referee #1**

We appreciate the feedback provided by the anonymous reviewer. We respond by highlighting the reviewer's comment in shaded text and our responses in black. The resulting changes from our responses to referee #1 are highlighted in blue in the revised manuscript.

**Comments:**

1. The authors have done a commendable job in addressing my comments in the revised submission and the overall quality of the presentation has improved substantially. I have one minor suggestions below.

Thank you for your positive feedback and recognition of the improvements in our revised submission. We are pleased that the additional tests we conducted based on your previous comments have enriched the manuscript and bolstered the robustness of our results.

2. L306-314 and Appendix B: The corresponding results that demonstrate the robustness of the choice of parameters to process S/N as the authors presented in the "response to reviewers" should be summarized and included in the paper as well.

Following the reviewer suggestion, we add more details about the sensitivity tests that guided our choices in Appendix B and section "Discussion and conclusions".

Appendix B now includes one of the test figures and the text was modified as follows:

L415-419: "Our estimation of the S/N ratio follows the method proposed by Hawkins and Sutton (2012), with a thorough parameter selection process that includes sensitivity tests, which guided the rationale for our choices. The key sensitivity test includes the evaluation of different filters, window types, and window sizes (as exemplified in Fig. B1). The tested filters are Butterworth, Chebyshev, Elliptic, FFT low-pass, and Lowess. The window types are Rectangular, Hamming, Bartlett, Blackman, and Hanning. The window lengths are 21, 31, 41, 51, and 61 years. The key findings are summarized as follows:"

L427-434: "… Filters that produce phase distortion can exaggerate differences between the original and filtered time-series, leading to a misrepresentation of the noise term with a higher value. For example, N ranges from 1.18 to 1.23 for filters that introduce phase distortion, whereas N=1.04 for the Lowess filter (Fig. B1a). Similarly, the "blending" of the FFT smoothed signal on the edges of the time-series may make the signal to unrealistically emerge or immerse on the natural variability range. The type and length of the window have a relatively minor impact in comparison. In summary, while we recognize that the selection of window length and smoothing options involves some subjectivity, the resulting N term exhibits relatively low sensitivity to reasonable choices, varying less than 9 % among the entire set of window types and lengths tested for the global signal (Fig. B1b-c). …"

L440-445: "… Lastly, alternative methods, such as estimating local signals by filtering local time-series, can produce misleading results. For instance, we tested this alternative for three rivers with different regional trends in their future mean flows: Congo (positive), Amazon (neutral), and Negro (negative). The results reveal that local filtered signal is affected by the filter's "blending" effects on extremes, regardless of regional trends. Moreover, this "blending" yields to high S/N ratio values in the first year of simulation, which is unrealistic. This issue is avoided when linear regressions are applied."

[Figure]

Figure B1. Low-pass filter selection test for the global river discharge anomaly time-series ($Q_G$) using the CNRM-CM6-1-HR model as a case study. The signal $S_G$ is estimated with various (a) filters, (b) window functions, and (c) window sizes. The (b) and (c) tests are applied for the Lowess filter. The resulting noise value of each filter is provided in the legend.

On the other hand, the paragraph associated to the filtering process in the section "Discussion and conclusions" now states:

L311-325: "The assessment of hydrological simulations involved estimating the S/N ratio and ToE to determine, for which rivers of the world, and when, the climate change signal will emerge from the natural variability. The method used for the calculation of the signal (both global and local) is key, as it determines the noise terms and the ToE. On one hand, the selection of filter, window function, and window length is key for estimating the global signal. Guided by rigorous sensitivity tests of various options, our analysis favored the use of a Lowess filter with a Hanning window of 41-year length. Our results indicated that the choice of filter significantly impacts the signal, as some filters introduce phase distortion and/or a "blending" effect, leading to erroneous high noise values that alter the ToE. These drawbacks are minimized by the Lowess filter, which effectively emphasizes long-term variations without introducing any distortion. Despite the subjectivity inherent in these choices, our analysis revealed that the resulting noise term exhibited low sensitivity to reasonable options of window function and length. A Hanning window of 41-year length was chosen as it effectively emphasizes long-term variations, essential for capturing climate change signals amidst natural variability. On the other hand, local signals can be estimated through linear regression of the global signal or by filtering local time-series. Both alternatives were tested for rivers with different regional trends (positive, neutral, and negative). The results revealed that the local filtered signals are affected by the filter's "blending" effects on extremes, regardless of regional trends. However, this issue was effectively mitigated when linear regressions were applied. These methodological considerations enhance the reliability of our S/N ratio estimation for assessing ToE."

**Responses to Referee #2**

We thank the new revision offered by the anonymous referee. We respond by highlighting the reviewer's comment in shaded text and our responses in black. The resulting changes from our responses to referee #2 are highlighted in green in the revised manuscript.

**Main Comment:**

1. The manuscript "River flow in the near future: a global perspective in the context of a high-emission climate change scenario" presents a study of the possible effects of global warming in a high-emission scenario on river flows over the next few decades using model intercomparison. Overall, the goal of the paper is clear, and the writing is well organized. The regional implications of the results are detailed.

We appreciate your recognition of the clarity and organization of our paper, as well as your suggestions for further clarification and improvement.

2. However, my main concern is the lack of comparison of the study's results with similar work (e.g., www.nature.com/articles/s44221-023-00030-7, www.nature.com/articles/s41597-022-01410-6, doi.org/10.5194/hess-21-4379-2017). It is hard to identify the novelty of the study without comparison. Therefore, I would suggest that in the discussion section, instead of focusing on the regional implications again (as some are already stated in Section 3.4), the authors could make a comprehensive comparison of the results with the literature and indicate and interpret the possible disagreement and how it may be related to the methodology used, which may be more valuable for readers.

Thanks for your comment. In response to the lack of comparison with other studies, and the need to reinforce the novelty of our research, we have expanded the Discussion section with the following text (L384-398): "Recent studies reveal significant advancements in understanding the complex interplay between climate change and river flow projections. Bosmans et al. (2022) introduced a high-resolution dataset projecting global river flow under various climate scenarios, which resembles the patterns observed in our findings. Zhou et al. (2023), in agreement with our projections, attribute changes in runoff to shifts in land surface characteristics such as vegetation and soil conditions. Zhang et al. (2023) predict similar river flow anomaly patterns to those found in our work but suggest that global river flow may be lower than projected by GCMs, attributing this discrepancy to the heightened sensitivity of river flow to changes in evapotranspiration, linked to the phenomenon of vegetation greening. Our work complements these studies by employing advanced techniques such as the S/N ratio and the ToE, which are key for identifying when river systems may exhibit conditions beyond their known historical variability. Our findings underscore the pressing need for a paradigm shift in prioritizing water-related concerns in the context of climate change, as emphasized by Douville et al. (2022). Moreover, our study emphasizes the interplay between water cycle alterations and potential hydrological impacts, providing valuable insights for planning purposes. It is concerning that several major rivers are projected to

imminently surpass the bounds of their natural variability. However, the hydrological predictions presented in this work should be interpreted in the context of a very high baseline emission scenario, i.e., an outcome only likely if society does not make concerted efforts to reduce greenhouse gas emissions (Van Vuuren et al., 2011). In future work, we will extend the analysis to encompass a broader set of the new SSP scenarios."

We would like to clarify that while Section 3.4 evaluates model projections in regions with strong agreement, the Discussion section delves into real-world consequences (recent observed and potential future impacts) in such regions. Both sections offer distinct perspectives on the topic.

Lastly, we respectfully argue against comparing with Papadimitrious et al. 2017. Their research focuses on biases of runoff generated with uncoupled land surface model simulations forced by GCM atmospheric variables, which do not account for feedback from the land surface to the atmosphere, thereby introducing uncertainty into the simulated runoff fields. In contrast, our study we use the simulated runoff from couple GCMs simulations to force a routing model, which transports the runoff from its origin to the river channels, without altering the land-atmosphere water balance. This fundamental difference precludes extending their findings about runoff biases to our simulations, a point deeply evaluated in Section 3.1 of our manuscript.

Bosmans, J., Wanders, N., Bierkens, M. F., Huijbregts, M. A., Schipper, A. M., and Barbarossa, V. (2022). FutureStreams, a global dataset of future streamflow and water temperature. *Scientific data*, 9(1), 307.

Papadimitriou, L. V., Koutroulis, A. G., Grillakis, M. G., and Tsanis, I. K. (2017). The effect of GCM biases on global runoff simulations of a land surface model. *HESS*, *21*(9), 4379-4401.

Van Vuuren, D. P., Edmonds, J., Kainuma, M., Riahi, K., Thomson, A., Hibbard, K., Hurtt, G., Kram, T., Krey, V., Lamarque, J.-F, Masui, T., Meinshausen M., Nakicenovic, N., Smith, S., and Rose, S. K. (2011). The representative concentration pathways: an overview. *Climatic change*, *109*, 5-31.

Zhang, Y., Zheng, H., Zhang, X., Leung, L. R., Liu, C., Zheng, C., Guo, Y., Chiew, F., Post, D., Kong, D., Beck, H., Li, C., and Blöschl, G. (2023). Future global streamflow declines are probably more severe than previously estimated. *Nature Water*, 1(3), 261-271.

Zhou, S., Yu, B., Lintner, B. R., Findell, K. L., and Zhang, Y. (2023). Projected increase in global runoff dominated by land surface changes. *Nature Climate Change*, *13*(5), 442-449.

**Minor Comments:**

1. L24: In some places, ET has been shown to be more influenced by vegetation change (e.g., https://www.nature.com/articles/s43017-023-00464-3).

Thanks for noting it. This is now reflected in L24-L26:

"… Evapotranspiration has changed in response to changes in precipitation and warmer temperatures, as well as to the observed vegetation greening in northern high latitudes (Yang et al. 2023), altering the ability of the soil to hold moisture. …"

Yang, Y., Roderick, M. L., Guo, H., Miralles, D. G., Zhang, L., Fatichi, S., Luo, Y., McVicar, T., Tu, Z., Keenan, T., Fisher, J., Gan, R., Zhang, X., Piao, S., Zhang, B., and Yang, D. (2023). Evapotranspiration on a greening Earth. *Nature Reviews Earth & Environment*, 4(9), 626-641.

2. L38: It would be beneficial if the author could clarify the difference between "runoff" and "streamflow" here.

The sentence now states: "This uncertainty extends to runoff, and by that to river flow, which is the local runoff that is subsequently routed from land to oceans through river channels." (see L39). Note that "streamflow" has been replaced by "river flow" (see our response to comment #4).

3. L39, It might be worth mentioning https://doi.org/10.1038/s41558-023-01659-8 here to indicate the role of land surface change as well.

We discuss the suggested paper in the revised manuscript (L39-L44): "Douville et al. (2021) and Zhou et al. (2023) concur on the projected increase in global runoff in the coming decades, albeit attributing it to different factors. Douville et al. (2021) link this rise to global warming, with confidence levels escalating with emissions scenarios. In contrast, Zhou et al. (2023) attribute it to changes in the synergistic effects of vegetation responses to rising $CO_2$ concentrations and land surface reactions to radiative changes, which lead to a shift in precipitation partitioning towards runoff instead of evapotranspiration."

4. L42, Does "river flow" have the same meaning as the previously used "streamflow"? Please consider making the wording consistent in case of confusion.

Effectively, river flow has the same meaning than streamflow. To ensure clarity and consistency, we have replaced all instances of 'streamflow' with 'river flow' throughout the manuscript (L39, L134, L265, 295, 485).

5. L56, "Hydrological model" is confusing here, as it seems to indicate only a stand-alone routing model.

The term "hydrological model" has been replaced by "river routing model" (L59).

6. L71-73, It is hard for me to connect here the provision of resolution (I assume it is by Table 1) and the reconciliation of variability in network topologies. I think the authors wanted to point out that only 50N information is shown because of the different network topologies used.

The reviewer is correct in the understanding. We have clarified it as follows (L76-L80): "To reconcile the variety of grid topologies used by the different GCMs (rectilinear, reduced gaussian, icosahedral, etc) in the comparison of the GCMs' resolution, we provide the atmospheric horizontal resolution at 50 ° N. This mid-latitude serves as a representative point for assessing

resolution, particularly given the significant variation in resolution from the Equator to the poles in models using rectilinear grids. The atmospheric resolution at 50 ° N ranges from 25 km to 134 km for the set of simulations (Table 1)."

Note that presenting the GCMs' atmospheric resolution at 50 ° N is a common practice in papers based on HighResMIP simulations (e.g. Vannière et al. 2019, Demory et al. 2020, Müller et al. 2021, among others).

Vannière, B., Demory, M. E., Vidale, P. L., Schiemann, R., Roberts, M. J., Roberts, C. D., Matsueda, M., Terray, L., Koenigk, T., and Senan, R. (2019). Multi-model evaluation of the sensitivity of the global energy budget and hydrological cycle to resolution. Climate dynamics, 52, 6817-6846.

Demory, M.-E., Berthou, S., Fernández, J., Sørland, S. L., Brogli, R., Roberts, M. J., Beyerle, U., Seddon, J., Haarsma, R., Schär, C., Buonomo, E., Christensen, O. B., Ciarlo`, J. M., Fealy, R., Nikulin, G., Peano, D., Putrasahan, D., Roberts, C. D., Senan, R., Steger, C., Teichmann, C., and Vautard, R. (2020). European daily precipitation according to EURO-CORDEX regional climate models (RCMs) and high-resolution global climate models (GCMs) from the High-Resolution Model Intercomparison Project (HighResMIP). *Geosci. Model Dev.,* 13, 5485–5506.

Müller, O. V., Vidale, P. L., Vannière, B., Schiemann, R., Senan, R., Haarsma, R. J., and Jungclaus, J. H. (2021). Land–atmosphere coupling sensitivity to GCMs resolution: A multimodel assessment of local and remote processes in the Sahel hot spot. *Journal of Climate,* 34(3), 967-985.

7. L87, it might be useful to explain the basic principle of the method (not necessarily with equations).

The new version of the manuscript includes a brief explanation of how the routing model works (L93-L97): "TRIPpy employs a simple advection method within a water balance model to route total runoff through the topography. This method calculates changes in river channel storage within each grid cell by accounting for the difference between the inflow, which includes both local runoff and contributions from upstream grid cells, and the outflow. The outflow is estimated using a linear function of storage, considering the river flow velocity and the river length between two connected grid cells. Detailed TRIPpy equations can be found in the appendix of Müller et al. (2021a), where..."

8. L106, Why would the range be (-inf, inf) here? Assuming all simulated flows are zero, the percentage difference would be -100% at most, or did I misunderstand the percentage difference?

Thank you for pointing out the error. Indeed, the valid range is [-100, inf). This has been corrected in the revised manuscript (L117).

9. L108, it is not clear what would be the denominator in the calculation.

The Overlapping Coefficient (OC) is a statistical measure used to compare the distributions of observed and simulated river flow time-series. It quantifies the degree of overlap between the probability distributions of the two datasets. The OC score is calculated by summing the minimum values of the relative frequencies of observations and simulations for each bin in the histograms

of the respective datasets (see equation 5 in Müller et al. 2021). Essentially, it evaluates how much the histograms overlap, with a score of zero indicating no overlap and a score of one indicating complete overlap. The denominator $N$ in the OC calculation is a fixed number which represents the total number of monthly observations for the monitored river, providing the length of the time-series for normalization purposes. We added an interpretation of the score in L119-120.

Müller, O. V., Vidale, P. L., Vannière, B., Schiemann, R., and McGuire, P. C. (2021). Does the HadGEM3-GC3. 1 GCM overestimate land precipitation at high resolution? A constraint based on observed river discharge. *Journal of Hydrometeorology*, *22*(8), 2131-2151.

10. L130, please specify the window width used in the study.

The sentence has been rephrased to clarify this as follows "The filter is based on the convolution of a scaled window of 41-year length with the signal, resulting in a smoothing effect of the inter-annual variability." (L140). Please note that the rationale behind this and other choices related to the filtering processes are explained in detail in Appendix B (as indicated in L147).

11. L156, I would be cautious about calling the consistency "remarkable."

The sentence has been rephrased to "Despite the diversity of these metrics, there is consistency across models, with all exhibiting values in a narrow range." in L166-L167. This modification helps soften the emphasis while still effectively conveying the idea of consistency among models.

12. L161, How is the -1.8% derived? It seems to be an average value, but it is not quite convincing for me to calculate only the algorithmic mean given the positive and negative biases present (may use the mean of the absolute values).

Thank you for your feedback. First, it's important to clarify that we mistakenly wrote -1.8% in the text, whereas the correct value is -1.5%, as accurately depicted in Fig. 1c. The -1.5% relative bias (RB) represents the difference between the river flow of the ensemble mean simulation and the observed river flow in the selected 20 largest monitored catchments. The ensemble mean simulation, being an average of simulations with varying biases (positives and negatives), inherently provides a more stable estimate of model performance, reducing the impact of outliers or biases in any single simulation. This advantage of ensemble mean simulations elucidates the RB value of -1.5%. On the other hand, our choice of relative bias over other percentage scores based on absolute differences (e.g., percentage absolute bias) serves a purpose, indicating whether simulations tend to systematically produce anomalous negative or positive river flows. This information, not provided by the other selected scores, is crucial for ensuring accurate historical river flow reproduction before trusting projections for the future.

Based on the previous explanation, we applied the following changes in the revised manuscript:

a) The value -1.8% was replaced by -1.5% in L172.
b) The term "ensemble mean" is now presented as "ensemble mean simulation" in L109, L167, L171, and in the caption of Figure 1.

c) The analysis of scores for the ensemble mean simulation is now explained as follows in L166-L169: "… Despite the diversity of these metrics, there is consistency across models, with all exhibiting values in a narrow range. The ensemble mean simulation, being an average of simulations with varying biases, inherently provides a more stable estimate of model performance, reducing the impact of outliers or biases in any single simulation. This advantage of the ensemble mean simulation explain why it outperforms most individual models, with RB=6.8%, OC=0.63, r=0.72, and npKGE=0.58. Among the top-performing models are the GCMs of the EC-Earth3P and MRI-AGCM3-2 families."

d) The description of RB now includes its interpretation justifying its choice (L116-L118): "Relative Bias (RB): Measures the percentage difference between total simulated and observed mean flow for all monitored rivers, indicating whether simulations tend to overestimate or underestimate river flows. Range: [-100, inf), Perfect score: 0."

13. L167, The analysis in the paragraph is a bit disjointed from the previous paragraphs; is there some intention to compare the resolution of the model with the performance? If so, what about other models?

Thank you for drawing attention to the disjointed nature of the paragraph in question. After thorough consideration, we have opted to remove it from the manuscript. While our original intention was to reconcile our current findings with those of our previous study (Müller et al. 2021), which focused on models' resolution, we recognize that this comparison may disrupt the flow of the section and could potentially divert the attention of readers from the focus of this paper.

Müller, O. V., Vidale, P. L., Vannière, B., Schiemann, R., and McGuire, P. C. (2021). Does the HadGEM3-GC3. 1 GCM overestimate land precipitation at high resolution? A constraint based on observed river discharge. *Journal of Hydrometeorology*, *22*(8), 2131-2151.

14. Fig 1a) It may be more informative to use one of the four metrics instead of catchment size as the color.

Thank you for your suggestion. We have updated Figure 1a to reflect the overlapping coefficient (OC) of the ensemble mean simulation for each monitored river, as you recommended. The caption has also been revised accordingly.

[Figure]

(a) 346 monitored catchments

"Figure 1: Validation of the global hydrological simulations. (a) Monitored rivers with black dots indicating the observation sites for river flow and colors showing the overlapping coefficient (OC) of the ensemble mean simulation for each monitored river. (b) …"

15. L193, what does the ITCZ mean here?

ITCZ is the abbreviation of Intertropical Convergence Zone. It is now specified in its first occurrence (L197).

16. L316, should "increased" here be "increased precipitation"?

Thank you for noting the missing word "precipitation". It has been corrected in L327.